# Manipulation Intention Understanding for Accurate Zero-Shot Composed Image Retrieval

## Abstract

Composed Image Retrieval (CIR) facilitates retrieving an image matching a reference image while incorporating specified textual modifications, which is crucial for internet searches and e-commerce. Traditional supervised CIR methods rely on annotated triplets, which are labor-intensive and limit generalizability. Recent advances in Zero-Shot Composed Image Retrieval (ZS-CIR) address the challenge of performing this task without annotated triplets. A key challenge in ZS-CIR is training models on limited intention-relevant datasets to understand human intention implicitly expressed in textual modifications for accurately retrieving target images. In this paper, we introduce an image-text dataset incorporated with pseudo-manipulation intentions to enhance the training of ZS-CIR models in understanding human manipulation intents. Based on our dataset, we propose a novel framework, De-MINDS, for capturing the intent humans aim to modify, thereby enhancing the ZS-CIR model's ability to understand human manipulation descriptions. Specifically, a simple mapping network first maps image information into language space and forms a target description with a manipulation description. Subsequently, De-MINDS captures intention-relevant information from target descriptions and converts them into several pseudo-word tokens for accurate ZS-CIR. The De-MINDS model exhibits robust generalization and significant improvements in performance across four ZS-CIR tasks. It achieves performance improvements from 2.05% to 4.35% over the best methods and establishes new state-of-the-art results with comparable inference times. Our code is available at `https://anonymous.4open.science/r/De-MINDS/`.

## 1 Introduction

Composed Image Retrieval (CIR) [55] aims to retrieve an image that is visually similar to a reference image while having visual modification according to the manipulation text. Different from traditional image retrieval [15], CIR offers more flexibility and accuracy by enabling users to integrate both visual and textual information into their search intent. This approach has gained emerging attention in internet searches and e-commerce applications [12, 45]. Various supervised methods have been proposed to solve CIR problem [12, 33, 19, 4], which requires a large amount of annotated triplets, *i.e.,* a reference image, a manipulated description, and a target image, for training task-specific retrieval models. However, these supervised methods are labor-intensive for data annotation and tend to suffer from limited generalization capabilities due to bias in human annotation. To enhance model generalization and perform CIR tasks without annotated triplets, recent research [45, 3, 52, 25, 20] introduce Zero-Shot Composed Image Retrieval (ZS-CIR). Existing solutions for ZS-CIR map an image to the language space, combining it with text to form a query. This query retrieves target images from the shared semantic space of a pre-trained vision-language model by calculating semantic similarity. These methods typically involve a pre-trained mapping network that converts the reference

Submitted to 38th Conference on Neural Information Processing Systems (NeurIPS 2024). Do not distribute.

image into a pseudo-word token $S_*$. During retrieval, this token $S_*$ is merged with the manipulation description to construct a target description, which a pre-trained CLIP model [41] then encodes, leveraging its comprehensive pre-trained knowledge across image candidates for retrieval.

Despite remarkable advancement, the pre-trained mapping networks are not satisfactory for CIR due to the following reasons:

(1) There exists a discrepancy between the retrieval and pre-training stages in ZS-CIR models. During retrieval, the mapping network is tasked with aligning intent-specific visual information (*e.g.,* objects, scenes, colors, and styles) in language space to form a composed image description query (*e.g.,* change to a man playing the accordion joyfully in the street) for calculating semantic similarity with the target image. However, in the pre-training phase, the mapping network aligns general visual information with textual descriptions of the image content (*e.g.,* a musician plays the piano). Without intent-specific mapping, the pseudo-token $S_*$ contains heavy information redundancy involving most objects, background/foreground, color, and style, leading to inaccurate retrieval.

(2) Accurately understanding the intention a user intends to modify in manipulation descriptions presents substantial challenges. These intentions are implicitly expressed in users' manipulation descriptions. For instance, the manipulation intention embedded in the request to "make this photo feel like early fall" may involve changing colors (*e.g.,* orange and yellow), adjusting the scene (*e.g.,* fallen leaves), and adding specific objects (*e.g.,* autumnal trees). However, existing ZS-CIR models rely on the CLIP language encoder, which challenges capturing fine-grained/long information from text [51, 58], facing difficulties in accurately understanding these manipulation intentions.

In this work, we introduce the intent-CC3M, an intention-based dataset for training mapping networks capable of aligning intention-relevant visual information within the language space, thus addressing the gap between pre-training and retrieval in ZS-CIR models. We incorporate pseudo-manipulation descriptions in CC3M [47], the widely used ZS-CIR training dataset [45, 52]. These pseudo descriptions, reflecting potential user intention to manipulate images, are reasoned through chain-of-thought prompting using an off-the-shelf Multi-modal Large Language Model (MLLM), facilitating the learning of intent-specific mapping capabilities. Furthermore, to overcome the challenge of existing ZS-CIR models in understanding manipulation intention within descriptions, we propose a novel *unDErstanding of Manipulation INtention from target Description before Searching* approach, named De-MINDS. We leverage pseudo-manipulation descriptions to train De-MINDS to capture manipulation intention from various aspects (*e.g.,* objects, scenes, colors, styles) guided by multiple learnable queries. This intention information is mapped to several pseudo-word tokens, which are subsequently input into the CLIP language encoder, enhancing its ability to understand users' intention to modify and thereby improving the accuracy of CIR.

The main contributions of this work are summarized as follows: (1) We introduce intent-CC3M, a novel dataset with pseudo-manipulation descriptions reasoned by an MLLM to bridge the gap between pre-training and retrieval in ZS-CIR models. Our experiments demonstrate that baseline models trained with our dataset are capable of aligning intention-relevant visual information, achieving consistent performance improvements. (2) We propose a novel manipulation intention understanding network. We extract intentions in manipulation descriptions under the guidance of learnable queries and map to several pseudo-word tokens for retrieval, enhancing the CLIP's ability to understand users' intentions. It sheds new light on intention-based image retrieval. (3) Our De-MINDS are consistently effective and generalizable across diverse ZS-CIR tasks. It significantly improves CIR performance from 2.05% to 4.35% across four CIR tasks, establishing new state-of-the-art results with comparable inference time, further impacting vision and language applications.

## 2  Related Works

**Composed Image Retrieval.** Composed Image Retrieval (CIR) integrates image and text for retrieval [54]. Current models typically employ late fusion for integrating visual and language features separately [4, 33, 4]. In contrast, zero-shot CIR models like Pic2Word [45], SEARLE [3], and Context-I2W [52] train on image-text pairs, bypassing the need for costly CIR datasets. Pic2Word aligns entire images into text features, SEARLE adds a pseudo-word token to GPT-based captions, and Context-I2W employs context-dependent word mapping for accurate retrieval. However, these methods rely on the pre-trained CLIP language encoder, which struggles to understand intentions within manipulation descriptions. To tackle this issue, we propose a novel model that effectively

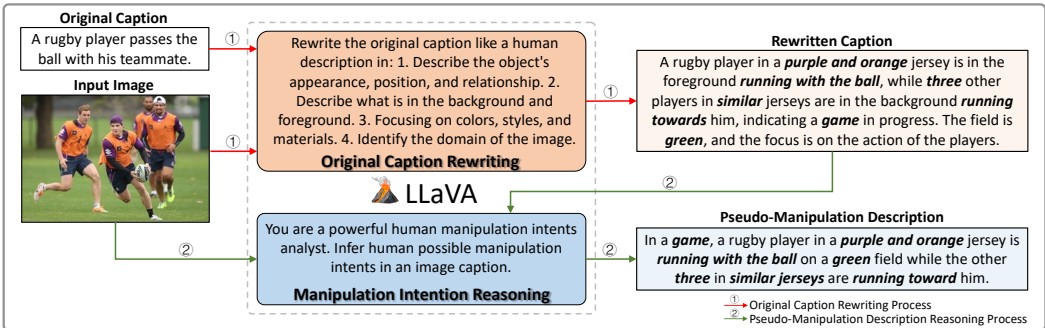

Figure 1: Illustration of using LLaVA to create our intent-CC3M dataset. We first use a prompt to guide the LLaVA model in generating rewritten captions with multi-view visual descriptions. Then, we leverage another prompt to reason pseudo-manipulation descriptions with potential intentions.

understands these intentions, thereby improving the ZS-CIR model's ability to retrieve images based on human manipulation intents accurately. Unlike CIReVL [25], which employs LLMs during inference for composed retrieval, introducing non-negligible computational overhead, our model is lightweight and achieves comparable inference time to recent approaches.

**Vision and Language Pre-training Models.** Vision and Language Pre-training (VLP) models, like CLIP [41], leverage extensive image-text pair training to achieve implicit alignment. Recent VLP advancements [60, 49] utilize static models to integrate encoded image and text features, enabling various zero-shot tasks [29, 49, 48]. However, current CLIP-based zero-shot learning struggles with manipulation description in CIR tasks, motivating our approach, which enhances CLIP's capabilities of understanding user intentions to modify from fine-grained/long descriptions. Moreover, recent studies [1, 28, 38, 37], inspired by DETR [7], employ learnable queries to select image and text information. In our work, we utilize multiple learnable queries to guide the extraction of manipulation intentions from target descriptions, providing explanatory cues for more accurate ZS-CIR.

**Image-text Dataset Enhancement.** In the field of vision-language learning, various endeavors [17, 27, 18, 39, 10] aim to enhance caption quality within existing image-text datasets. LaCLIP [17] utilizes LLMs to refine raw captions. VeCLIP [27] integrates insights from raw and synthetic sources using LLMs. The latest approach, ShareGPT4V [10], leverages MLLMs to generate descriptive captions from deliberate prompts and corresponding image inputs. However, these methods ignore human manipulation intentions, which are crucial for CIR tasks. To bridge this gap, we introduce a novel dataset infused with pseudo-manipulation intentions reasoned by MLLMs.

## 3 Methodology

### 3.1 Preliminary

Given a reference image space $\mathcal{I}$ and a text description space $\mathcal{T}$, Composed Image Retrieval (CIR) involves a user manipulation text $T \in \mathcal{T}$ describing hypothetical semantic changes to a reference image $I_r \in \mathcal{I}$, aiming to retrieve a target image with its closest context from an image database $\mathcal{D} = \{I_i, \ldots, I_n\}$. Zero-Shot CIR (ZS-CIR) approaches [45, 3, 52] sidestep this requirement by training a mapping network to map the reference image into an associated text representation. Specifically, these methods learn a mapping function $f_\theta : \mathcal{I} \to \mathcal{Z}$, where $\mathcal{Z}$ is a pre-defined text-token embedding space. $f_\theta$ is trained using intermediate image representations from a specific image encoder $\Psi_I$, often part of a pre-trained vision-language representation system. Template filling around the manipulation text over the pseudo token embedding $\boldsymbol{S}_* = f_\theta(\Psi_I(I_r))$ is then employed to aggregate information into a target description $P$ (*e.g.,* "a photo of $S_*$, $\{T\}$)." This target description serves as input for target image retrieval, encoding it using the associated pre-trained text encoder $\Psi_T$. The respective matching score is $\texttt{cos\_sim}(\Psi_I(I_r), \Psi_T(P))$ using cosine similarity.

### 3.2 Creating Intention-based Image-text Aliagment Dataset

To address the discrepancy between pre-training and retrieval in existing ZS-CIR models, we aim to develop an intention-based image-text dataset for training mapping networks capable of aligning

intent-relevant visual information within the language space. To make a fair comparison and mitigate the bias in human annotation, we propose to augment the widely used ZS-CIR training image-text dataset, CC3M, through LLaVA [32], an open-source, state-of-the-art Multi-modal Large Language Model (MLLM) known for its robust performance in vision-language tasks. However, reasoning potential manipulation intentions from image-text pairs remains a challenging task for LLaVA.

Recent advancements in MLLMs include the development of Chain-of-Thought (CoT) prompting [56], which enables MLLMs to produce a sequence of reasoning steps, breaking down multi-step problems into intermediate stages and enhancing performance in complex tasks [24]. Inspired by the CoT prompting mechanism, we explore a novel multimodal CoT prompting strategy using LLaVA to reason pseudo-manipulation descriptions with potential intentions from image-text pairs effectively.

As illustrated in Figure 1, we divide the process of reasoning pseudo-manipulation descriptions into two stages: the *Caption Rewriting* stage rewrites the original caption with multi-view visual information for CIR tasks. The *Intention Reasoning* stage further understands the manipulation intentions from rewritten captions to reason pseudo-manipulation descriptions. Specifically, in the caption rewriting stage, we utilize the $i$-th image $I_i$ and its original caption $T_{ori}^i$ from the CC3M, denoted as $\mathcal{D} = \{(I_r^i, T_{ori}^i), \ldots, (I_r^n, T_{ori}^n)\}$. We guide the LLaVA model with a prompt to generate a rewritten caption $T_{rew}^i$ for each image. These rewritten captions, averaging 65 tokens, include various aspects of visual information (*e.g.,* object, foreground/background, color, and domain style). In the intention reasoning stage, we apply an additional prompt to reason manipulation intention for rewritten captions. This results in a more effective pseudo-manipulation description $T_{int}^i$, averaging 27 tokens. The result dataset is represented as $\tilde{\mathcal{D}} = \{(I_r^i, T_{ori}^i, T_{rew}^i, T_{int}^i), \ldots, (I_r^n, T_{ori}^n, T_{rew}^n, T_{int}^n)\}$.

### 3.3 Manipulation Intention Understanding From Descriptions Before Searching

Since ZS-CIR models leverage the CLIP language encoder, there is a challenge in understanding manipulation intentions that are implicitly expressed in user descriptions. To address this challenge, we propose a method to understand the manipulation intention before feeding into the CLIP language encoder for accurate ZS-CIR in two modules: the *Manipulation Intention Understanding* captures manipulation intentions and maps them into several pseudo tokens. The *Reasoning Distillation* further aligns the context of desired pseudo-word tokens closely with human intention by leveraging pseudo-manipulation description to enhance the models' ability to understand human intention.

**Image and Context Encoding.** For a given sample $(I_r, T_{ori}, T_{rew}, T_{int})$ from intent-CC3M. Since the pre-trained vision-language models are strong at modeling the cross-modal implicit alignment. Initially, we employ the frozen image encoder $\Psi_I$ from the CLIP model to encode the global image feature of the reference image $I_r$ as $\boldsymbol{v} = \Psi_I(I_r) = \{v_i\}_{i=1}^d \in \mathbb{R}^{d \times 1}$. Subsequently, we apply a simple mapping network $f_\theta$ with parameters $\theta$ to extract a pseudo token embedding $S_* = f_\theta(\boldsymbol{v})$. Considering our focus on manipulation intention understanding for ZS-CIR, $f_\theta$ is structured as a simple three-layer fully-connected network. We then construct a target description $P$ formatted as "a photo of $S_*$, $\{T\}$". We consider two scenarios for manipulation intention understanding: deducing intention information from concise texts (*e.g.,* original caption) or integrating it from lengthy texts(**e.g.,** rewritten caption). Accordingly, the text $T$ is composed randomly within a batch according to the following distribution: 50% original caption $T_{rew}$ and 30% rewritten caption $T_{ori}$ to learn manipulation intention understanding, 20% pseudo-manipulation description $T_{int}$ to ensure training stability (details are in Appendix C). We feed the target description to the language encoder $\Psi_T$ of frozen CLIP to represent the target description $P$ by a set of language feature vectors $\boldsymbol{T}$ $=\{\boldsymbol{t}_i\}_{i=1}^m \subseteq \mathbb{R}^{d \times m}$. $\boldsymbol{t}_1$ represents the [CLS] embedding $\boldsymbol{t}_{cls}$ with global information of image and caption, while other ones denote word embeddings $\tilde{\boldsymbol{T}} = \{\boldsymbol{t}_i\}_{i=2}^m$.

**Manipulation Intentions Understanding.** Given the word embeddings of the target descriptions, this module aims to capture different manipulation intentions, thereby enhancing the CLIP language encoder's capability to understand users' intents for manipulation. To capture different manipulation intentions, we introduce a set of learnable query embeddings for guidance, denoted as $\boldsymbol{X} = \{\boldsymbol{x}_k\}_{k=1}^n \in \mathbb{R}^{d \times n}$, where $d$ is the embedding dimension and $n$ is the number of queries. Each query $\boldsymbol{x}_k$ represents a kind of manipulation intention. As depicted in Figure 2(left), we implement cross-attention mechanisms to extract intention-relevant contextual information from the word embeddings $\tilde{\boldsymbol{T}} = \{\boldsymbol{t}_i\}_{i=2}^m$ using the learnable queries $\boldsymbol{X}$. The cross-attention operation involves three primary steps. First, we compute the query, key and value through linear projections,

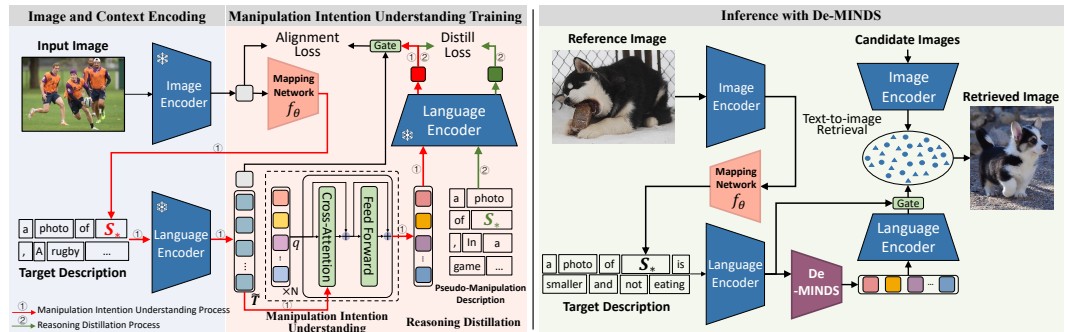

Figure 2: An overview of our De-MINDS. Pre-training (left): Map the image to a pseudo token $S_*$, and understand the intention from the target description. Inference (right): Map the inference image to $S_*$ to construct the target description and understand manipulation intention for ZS-CIR.

*i.e.*, $Q = XW^Q$, $K = [X, \tilde{T}]W^K$, $V = [X, \tilde{T}]W^V$. $[X, \tilde{T}]$ denotes concatenating the two matrices, which enhances the interaction between learnable queries and word embeddings with better performance. Then, the learnable queries from the current cross-attention block $X^i$ is calculated as:

$$X^i_{att} = \text{Att}(Q, K, V) = softmax\left(\frac{QK^\top}{\sqrt{d}}\right)V, X^i = \text{FFW}(X^i_{att} + X^{i-1}) + X^i_{att} \quad (1)$$

where $X^{i-1}$ are learnable queries from the previous block and $\text{FFW}(\cdot)$ denotes 2-layer feed-forward networks. the refined query embeddings $X$ are then fed into the frozen language encoder $\Psi_T$ of CLIP to extract the intention embedding as $t_* = \Psi_T(X^n) = \{t^i_*\}^d_{i=1} \in \mathbb{R}^{d \times 1}$ ($d = 768$).

**Reasoning Distillation.** Given the intention embedding $t_*$, the AI agent needs to further align with human manipulation intention. Specifically, we aim to reduce the distance between the intention embedding and the corresponding pseudo-manipulation description's [CLS] word embedding, which represents the MLLM's intention embedding while ensuring that each embedding remains distinct and discriminative. Given the intention embeddings $\mathcal{T}_{int} = \{t^i_*\}^N_{i=1}$, where $N$ is the number of images in $\tilde{D}$, and the corresponding MLLM's intention embeddings $\tilde{t}_* = \Psi_T(T_{int}) \in \tilde{\mathcal{T}}_{int}$ we employ a symmetric contrastive loss inspired by SimCLR [11, 13, 45] as follows:

$$\mathcal{L}_{distil} = \mathcal{L}_{s2t}(t_*, \tilde{t}_*) + \mathcal{L}_{t2s}(\tilde{t}_*, t_*) \quad (2)$$

The two contrastive loss terms are defined as:

$$\mathcal{L}_{s2t}(t_*, \tilde{t}_*) = -\frac{1}{|\mathcal{B}|}\sum_{i\in\mathcal{B}}\log\frac{e^{\tau(t^i_*)^T\tilde{t}^i_*}}{\sum_{j\in\mathcal{B}}e^{\tau(t^i_*)^T\tilde{t}^j_*}}, \mathcal{L}_{t2s}(\hat{t}_*, \tilde{t}_*) = -\frac{1}{|\mathcal{B}|}\sum_{i\in\mathcal{B}}\log\frac{e^{\tau(\tilde{t}^i_*)^T t^i_*}}{\sum_{j\in\mathcal{B}}e^{\tau(\tilde{t}^i_*)^T t^j_*}} \quad (3)$$

where $B$ is the number of images in a batch and $\tau$ is a temperature hyper-parameter that controls the strength of penalties on hard negative samples.

**Cross-Modal Alignment.** Given the embedding of user manipulation intention, this module aims to form a target embedding optimized for retrieval. Since the nature of CIR, both the reference image and the manipulation intention form a comprehensive context that defines the target image. To dynamically control the influence of manipulation intentions on the retrieval process, we introduce a learnable scalar $gate$ that decides the contribution of the manipulation intention information $t_*$ and integrates the global information $t_{cls}$ to form the final target embedding $\hat{t}$ as follows:

$$\hat{t} = t_{cls} + gate \cdot t_*$$

Then, we aim to match a target image to its paired target embedding while separating unpaired ones. We minimize the symmetric contrastive loss between the image embedding $v$ and the target embedding $\hat{t}$ as follows:

$$\mathcal{L}_{align} = \mathcal{L}_{s2t}(\hat{t}, v) + \mathcal{L}_{t2s}(v, \hat{t}) \quad (4)$$

where $\mathcal{L}_{s2t}$ and $\mathcal{L}_{t2s}$ are two contrastive loss terms as Eq.3. The final loss used to optimize is:

$$\mathcal{L} = \mathcal{L}_{distil} + \mathcal{L}_{align} \quad (5)$$

**Inference with De-MINDS.** In the inference stage, we compose the reference image with the paired manipulation description and compare the composed query with candidate images for retrieval. As shown in Figure 2 (right), we compose the pseudo token embedding $S_*$ of the image from the mapping network with the text description and feed it to the pre-trained language encoder of CLIP. The result is embedded by the text encoder and compared to the visual features of candidate images.

Since we focus on studying the manipulation intention understanding searching for ZS-CIR, we utilize the same prompt in the most recent works [45, 52] for a fair comparison. We show prompt examples for different ZS-CIR tasks. In all examples, [*] indicates the pseudo token from the mapping network: **(a) Domain conversion** aims to modify the domain of the reference image. The prompt is defined as a [domain tag] of [*]; **(b) Object composition** retrieves an image that contains an object in the reference image and other object tags. The prompt is in the format of a photo of [*], [obj$_1$ tag] and [obj$_2$ tag], ..., and [obj$_n$ tag]; **(c) Sentence manipulation** modifies the reference image based on a sentence. We simply append the sentence with the special token as a photo of [*], [sentence]. More details are in Appendix D.3.

# 4 Experiments

**Datasets.** We evaluate our model on four ZS-CIR datasets, *i.e.,* COCO [31] for object composition, ImageNet [16, 21] for domain conversion, CIRR [33] for object/scene manipulation, and Fashion-IQ [57] for attribute manipulation. All the dataset settings and evaluation metrics (Recall@K) follow the recent works [45, 52] for a fair comparison.

(1) Domain conversion. This dataset comprises 16,983 images of 200 classes from four domains, *i.e.,* cartoon, origami, toy, and sculpture. We use the prompt (a) in inference. (2) Object composition. The dataset contains images with corresponding lists of object labels and instance masks of query images. We randomly crop one object and mask its background using its instance mask to create a reference image. We use the prompt (b) in inference. (3) Object/scene manipulation. A reference image is an instruction for manipulating an object or the background scene. We apply the prompt (c) in inference. (4) Attribute manipulation. This dataset includes various description sentences for manipulating image attributes. We utilize the prompt (c) in inference. More details in Appendix D.2.

**Implementation Details.** Generating one pseudo-manipulation description through LLaVA-1.6-13B [32] for the entire Conceptual Caption dataset [47], which comprises 3M images (CC3M), requires approximately 625 hours on 5 A100 (80G) GPUs. For training De-MINDS, We utilize the CC3M and adopt ViT-L/14 CLIP [41] pre-trained on 400M image-text paired data. We employ AdamW [34] with a learning rate of $1 \times 10^{-6}$, weight decay of 0.1, and a linear warmup of 10000 steps. The number of cross-attention blocks is 6. The number of learnable queries is 4. The batch size for contrastive learning is 1024. To improve training stability, we initialize the learnable scalar of tanh-gating to 0 [2]. For training Context-I2W and SEARLE, we keep the same setting reported in their paper, only replacing the original captions with our pseudo-manipulation descriptions. All models are trained on 4 NVIDIA A100 (80G) GPUs. To ensure reliable results, we report the performance averaged over three trials. More details are in Appendix D.1.

## 4.1 Quantitative and Qualitative Results

We compare De-MINDS with several ZS-CIR methods, including: 1) **Pic2Word** [45]: Maps the visual features of a reference image into a pseudo-word token within the CLIP token embedding space; 2) **SEARLE-XL** [3]: Similar to Pic2Word, further integrating the pseudo-word token with the caption generated by GPT [6] and distilled for efficiency; 3) **Context-I2W** [52]: Selectively extracts text-relevant visual information from the reference image before mapping it into a pseudo-word token; 4) **CIReVL** [25]: Uses LLMs to enhance the manipulation description during inference; and 5) **LinCIR** [20]: Masks subjects in captions from various image-text datasets for training. For a fair comparison, we present the reported results of methods relying on the ViT-L/14 CLIP model.

Moreover, we compare De-MINDS with 6) **SEARLE-XL* and Context-I2W***: Replace the original captions with our pseudo-manipulation description, and standard ZS-CIR methods, including 7) **Text-only**: Computes similarity based on the CLIP features of descriptions and candidate images; 8) **Image-only**: Retrieves the most similar images to the reference image; and 9) **Image + Text**: Sums the CLIP features of the reference image and the description.

Table 1: Results on Fashion-IQ for attribute manipulation.

| Methods | Conferences | Dress | | Shrit | | TopTee | | Average | |
|---|---|---|---|---|---|---|---|---|---|
| | | R10 | R50 | R10 | R50 | R10 | R50 | R10 | R50 |
| Image-only | – | 5.4 | 13.9 | 9.9 | 20.8 | 8.3 | 17.7 | 7.9 | 17.5 |
| Text-only | – | 13.6 | 29.7 | 18.9 | 31.8 | 19.3 | 37.0 | 17.3 | 32.9 |
| Image+Text | – | 16.3 | 33.6 | 21.0 | 34.5 | 22.2 | 39.0 | 19.8 | 35.7 |
| Pic2Word [45] | CVPR 2023 | 20.0 | 40.2 | 26.2 | 43.6 | 27.9 | 47.4 | 24.7 | 43.7 |
| CIReVL [25] | ICLR 2024 | 24.6 | 44.8 | 29.5 | 47.4 | 31.4 | 53.7 | 28.6 | 48.6 |
| LinCIR [20] | CVPR 2024 | 20.9 | 42.4 | 29.1 | 46.8 | 28.8 | 50.2 | 26.3 | 46.5 |
| SEARLE-XL [3] | ICCV 2023 | 20.3 | 43.2 | 27.4 | 45.7 | 29.3 | 50.2 | 25.7 | 46.3 |
| SEARLE-XL* | – | 22.7 | 45.0 | 29.4 | 47.9 | 30.2 | 51.4 | 27.4 | 48.1 |
| Context-I2W [52] | AAAI 2024 | 23.1 | 45.3 | 29.7 | 48.6 | 30.6 | 52.9 | 27.8 | 48.9 |
| Context-I2W* | – | 23.9 | 46.9 | 30.4 | 49.7 | 31.1 | 53.8 | 28.5 | 50.1 |
| **De-MINDS** | – | **25.2** | **48.7** | **31.0** | **51.2** | **32.9** | **55.7** | **29.7** | **51.9** |

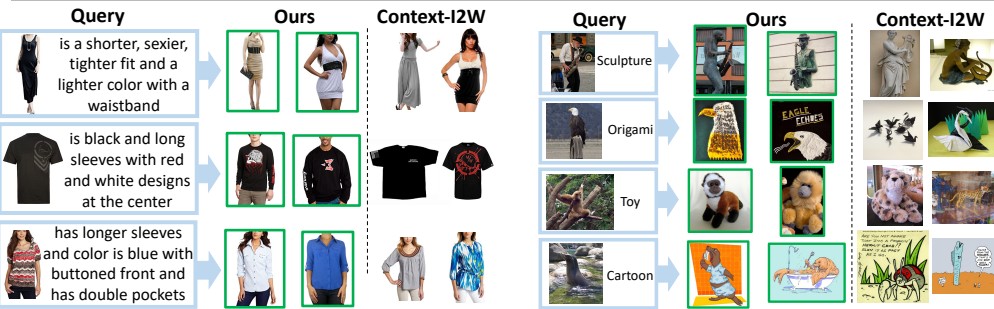

Figure 3: Results on the attribute manipulation task

Figure 4: Results on the domain conversion task.

Tables 1 to 4 present the quantitative results, while Figures 3 to 6 display the corresponding qualitative results of our model and the most recent works, CIReVL and Context-I2W. The attribute manipulation task requires accurately localizing specific attributes within the entire image. As demonstrated in Table 1, De-MINDS outperforms existing ZS-CIR models significantly, achieving an average improvement of 2.20% over the State-of-the-Art (SoTA) model, CIReVL. CIReVL's dependency on an LLM at inference introduces substantial computational overhead during retrieval. De-MINDS tackles this challenge by extracting fashion-relevant intention within manipulation descriptions into a series of implicit pseudo-tokens for CLIP retrieval. This approach is more efficient and suitable for models than relying on explicit, often noisy, LLM analysis results. Figure 3 further illustrates how De-MINDS effectively understand complex fashion-relevant attributes in manipulation descriptions, such as a sexier style with a waistband (row 1), black color with a special design in the center (row 2), and longer sleeves with two pockets in blue (row 3), facilitating more accurate searching.

We further assess De-MINDS' capability in foreground/background differentiation and fine-grained image editing through the object/scene manipulation task (Table 2). De-MINDS consistently surpasses existing ZS-CIR models, achieving an average performance improvement of 2.05% over the best model. This enhancement is attributed to De-MINDS' approach of extracting human intention from manipulation descriptions before searching, enhancing the ability of the CLIP language encoder to understand the user's intention to modify. In Figure 5, De-MINDS accurately understands manipulation intention to change the number of an object and modify the background (row 1), alter the stage and remove an overlapping object (row 2), adjust the camera focus, age of a dog, and remove a specific object (row 3), and modify the style of an image with a specific design (row 4).

In the object composition experiments (Table 3), De-MINDS significantly outperforms the current SoTA model by an average of 4.30%. These results prove the effectiveness of De-MINDS in accurately mapping visual information to the language token space via bridges the gap between pre-training and retrieval, which facilitates the combination of multiple objects, as shown in Figure 6.

Moreover, in the domain conversion results (Table 4), De-MINDS consistently outperforms existing approaches and notably surpasses the SoTA Context-I2W by an average of 4.35%. As illustrated in Figure 4, De-MINDS accurately maps objects within complex scenes (e.g., a saxophonist in the street, a bald eagle on wood, a monkey in the forest, and a sea lion in the water). In contrast, Context-I2W struggles to select the intention-relevant local visual features due to its reliance on image caption without intention, whereas our pseudo-manipulation descriptions are effectively addressed.

Table 2: Results on CIRR for object manipulation task.

| Methods | R1 | R5 | R10 | R50 |
|---------|-----|-----|-----|-----|
| Image-only | 7.4 | 23.6 | 34.0 | 57.4 |
| Text-only | 20.9 | 44.8 | 55.5 | 79.1 |
| Image+Text | 12.4 | 36.2 | 49.1 | 78.2 |
| Pic2Word [45] | 23.9 | 51.7 | 65.3 | 87.8 |
| CIReVL [25] | 24.6 | 52.3 | 64.9 | 86.3 |
| LinCIR [20] | 25.0 | 53.3 | 66.7 | – |
| SEARLE-XL [3] | 24.2 | 52.4 | 66.3 | 88.6 |
| SEARLE-XL* | 25.4 | 54.1 | 66.9 | 89.3 |
| Context-I2W [52] | 25.6 | 55.1 | 68.5 | 89.8 |
| Context-I2W* | 26.3 | 55.7 | 69.0 | 90.2 |
| **De-MINDS** | **27.3** | **57.0** | **71.3** | **91.6** |

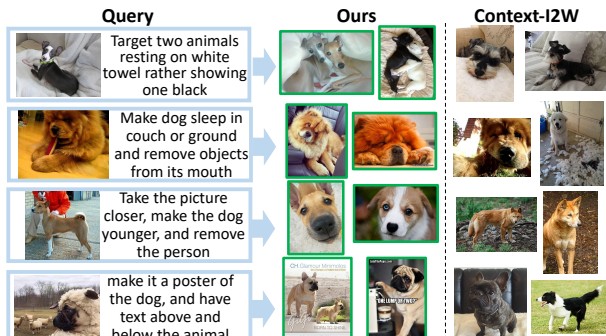

Figure 5: Retrieved results on the object manipulation task

Table 3: Results on COCO for object composition task.

| Methods | R1 | R5 | R10 |
|---------|-----|-----|-----|
| Image-only | 8.6 | 15.4 | 18.9 |
| Text-only | 6.1 | 15.7 | 23.5 |
| Image+Text | 10.2 | 20.2 | 26.6 |
| Pic2Word [45] | 11.5 | 24.8 | 33.4 |
| Context-I2W [52] | 13.5 | 28.5 | 38.1 |
| Context-I2W* | 14.3 | 29.7 | 40.5 |
| **De-MINDS** | **15.7** | **33.2** | **44.1** |

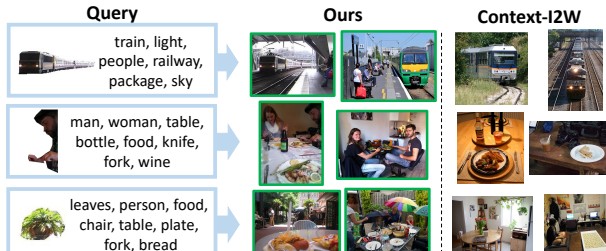

Figure 6: Retrieved results on the object composition task.

Table 4: Results on ImageNet for domain conversion.

| Methods | Conferences | Cartoon R10 | Cartoon R50 | Origami R10 | Origami R50 | Toy R10 | Toy R50 | Sculpture R10 | Sculpture R50 | Average R10 | Average R50 |
|---------|-------------|------|------|------|------|------|------|------|------|------|------|
| Image-only | – | 0.3 | 4.5 | 0.2 | 1.8 | 0.6 | 5.7 | 0.3 | 4.0 | 0.4 | 4.0 |
| Text-only | – | 0.2 | 1.1 | 0.8 | 3.7 | 0.8 | 2.4 | 0.4 | 2.0 | 0.5 | 2.3 |
| Image+Text | – | 2.2 | 13.3 | 2.0 | 10.3 | 1.2 | 9.7 | 1.6 | 11.6 | 1.7 | 11.2 |
| Pic2Word [45] | CVPR 2023 | 8.0 | 21.9 | 13.5 | 25.6 | 8.7 | 21.6 | 10.0 | 23.8 | 10.1 | 23.2 |
| Context-I2W [52] | AAAI 2024 | 10.2 | 26.1 | 17.5 | 28.7 | 11.6 | 27.4 | 12.1 | 28.2 | 12.9 | 27.6 |
| Context-I2W* | – | 11.2 | 27.4 | 18.7 | 30.4 | 12.5 | 29.8 | 13.7 | 31.4 | 14.0 | 29.8 |
| **De-MINDS** | – | **13.3** | **31.2** | **20.3** | **34.5** | **14.7** | **31.7** | **16.5** | **34.7** | **16.2** | **33.0** |

## 4.2 Ablation Study

In Table 5, we evaluate the contributions of De-MINDS components on the CIRR and FashionIQ datasets. **(1) In models '2-3', we assess the significance of the intent-CC3M dataset.** Replacing the pseudo-manipulation description with original captions (model '2') results in an average performance drop of 3.80%, demonstrating training with intent-CC3M benefit for aligning intention-relevant visual information. Using a single prompt for pseudo-manipulation descriptions (model '3') causes a 3.14% performance decline, indicating that CoT prompting enhances MLLM in reasoning potential manipulation intention. **(2) In models '4-6', we evaluate key modules in the manipulation intention understanding process.** Without intention embeddings from De-MINDS (model '4'), performance drops by 4.02% on average, proving De-MINDS's importance in CIR. Removing the global feature $t_{cls}$ (model '5') leads to a 2.38% performance decline, highlighting the necessity of comprehensive both global and intention information. Summing global and intention features directly (model '6') causes a 1.64% performance drop, indicating the need for adaptive capture of complementary information. **(3) In models '7-9', we assess De-MINDS's training strategies.** Using only original captions as $T$ (model '7') reduces training stability, resulting in a 1.62% performance drop. Without the distillation loss (model '8') or replacing it with a cosine loss (model '9') leads to performance drops of 3.58% and 1.54%, respectively, indicating the necessity of symmetric contrastive loss for distilling MLLM's reasoning ability. **In models '10-12', we evaluate alternative solutions.** Not utilizing $T$ for image-to-text mapping (model '10') results in a 2.30% performance drop, confirming the effectiveness of our pseudo-manipulation descriptions. Applying MiniGPT-4 [61] to generate the intent-CC3M dataset (model '11') results in a 1.18% performance drop, suggesting that a superior MLLM model benefits pseudo-manipulation description quality. Leveraging the LLaMA [53] rewrite

Table 5: Ablation study of main components on CIRR and FashionIQ.

| | | CIRR | | | Fashion-IQ | |
|---|---|---|---|---|---|---|
| | Methods | R1 | R5 | R10 | R10 | R50 |
| 1. | full model | 27.3 | 57.0 | 71.3 | 29.7 | 51.9 |
| **Significant of inetent-CC3M** | | | | | | |
| 2. | w/o intent-CC3M | 24.6 | 53.7 | 67.1 | 26.0 | 46.8 |
| 3. | w/o CoT | 25.2 | 54.3 | 67.8 | 26.7 | 47.5 |
| **Key modules of De-MINDS process** | | | | | | |
| 4. | w/o De-MINDS | 24.0 | 53.5 | 67.2 | 25.8 | 46.6 |
| 5. | w/o global feature | 25.5 | 55.2 | 68.0 | 27.3 | 49.6 |
| 6. | w/o gate | 25.9 | 55.3 | 69.5 | 27.9 | 50.4 |
| **Training Strategies** | | | | | | |
| 7. | w/o construct $T$ | 26.2 | 55.6 | 69.3 | 27.8 | 50.2 |
| 8. | w/o distil | 24.8 | 53.9 | 67.3 | 26.3 | 47.0 |
| 9. | cos distll | 26.2 | 55.5 | 69.7 | 27.9 | 50.2 |
| **Alternative solutions** | | | | | | |
| 10. | a photo of $S_*$ | 25.5 | 55.2 | 67.9 | 27.5 | 49.6 |
| 11. | MiniGPT4's caption | 26.4 | 55.7 | 70.2 | 28.2 | 50.8 |
| 12. | LLM's caption | 25.2 | 53.7 | 67.2 | 26.9 | 47.2 |

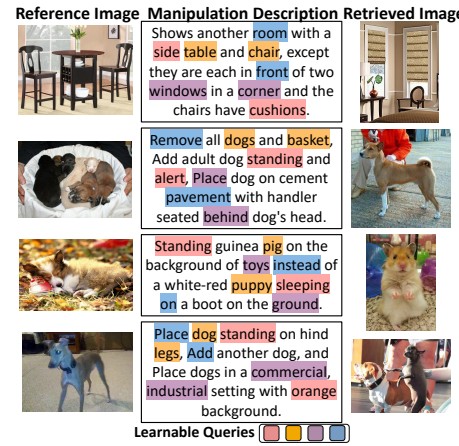

Figure 7: Visualization of the top two attention words for each learnable query, different colors denoting the results corresponding to each query.

CC3M dataset [17] (model '12') causes a 3.40% performance drop, indicating the necessity of MLLM for generating pseudo-manipulation description with multi-view supplementary image detail.

## 4.3 Analysis

**Interpretability of Learnable Query.** In Figure 7, we visualize the top two attention words of each learnable query from the last block, demonstrating the distinct focus of the four queries. Specifically, the first two queries mainly focus on object and attribute information, while the last two queries mostly consider foreground/background and relation information. These attention maps substantiate De-MINDS's interpretability in extracting specific intention across various descriptions, supporting the understanding of intention from manipulation descriptions.

**Effectiveness and Efficiency Analysis.** Our approach achieves significant improvements on four widely compared ZR-CIR tasks from 2.05% to 4.35% over the SoTA models. Designed for understanding manipulation intention, the model size of De-MINDS(58.5M) is larger than the simple 3-layer MLP mapping (0.9M) of Pic2Word. Consequently, our training time (20 hours) is 6 hours longer than Pic2Word under the same settings. Notably, our inference time (0.017s) is ×58 faster than CIReVL (∼ 1s), which uses LLM for inference, and only 0.005s slower than Pic2Word. It's worth noting that our model using just 50% of the pre-training data achieves comparable performance to SoTA models (details are in Appendix A.2).

**Limitation.** While the training process for De-MINDS does not introduce significant additional memory or computational overhead, generating pseudo-manipulation descriptions using MLLMs can be computationally intensive. Moreover, these pseudo descriptions are not filtered, potentially introducing irrelevant details that do not align with actual human manipulation intention. Our paper aims to bridge the gap between pre-training and retrieval in ZS-CIR models and introduce a novel framework to enhance the model's capability to understand user intention. Future work could explore more efficient methods to generate pseudo-manipulation descriptions while maintaining performance.

## 5 Conclusion

In this paper, we introduce intent-CC3M, an intention-based dataset featuring pseudo-manipulation descriptions reasoned through chain-of-thought prompting by an MLLM for training mapping networks to align intention-relevant visual information. Leveraging intent-CC3M, we propose a novel manipulation intention understanding network that employs learnable queries to enhance the models' capability to understand user intention from manipulation descriptions for accurate CIR. De-MINDS shows strong generalization ability and remarkably improves the best performance of existing approaches on four diverse ZS-CIR tasks with comparable inference times. Our work inspires intention-based image retrieval and impacts diverse vision and language applications.

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

## A  Extended Analysis

### A.1  Analysis of the number of learnable queries.

We conduct analysis on the number of learnable query embedding $\boldsymbol{X} = \{\boldsymbol{x}_k\}_{k=1}^n \in \mathbb{R}^{d \times n}$ as shown in Figure 8. We find that $n = 2$ results in not learning sufficient intentions for manipulation, but

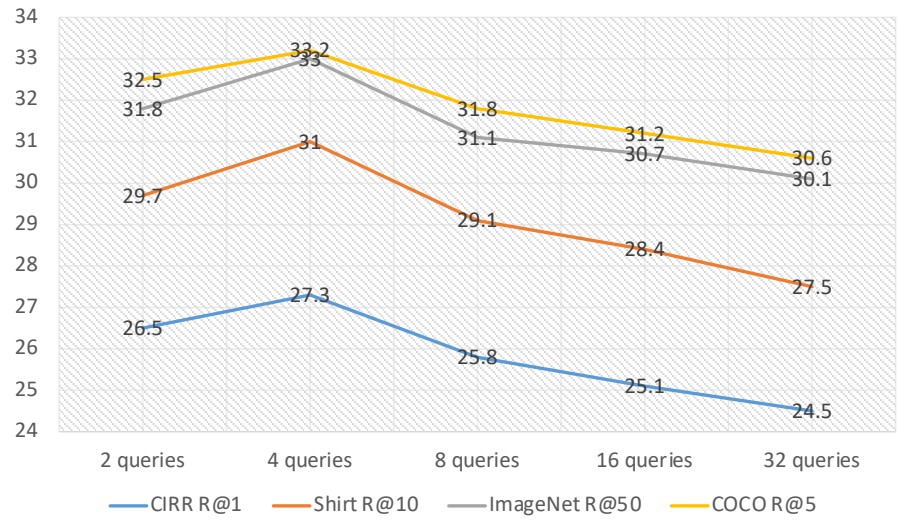

Figure 8: Analysis of the number of learnable queries.

when $n$ is added to 32, it is redundant and unhelpful for the CLIP model to understand manipulation intentions. We finally choose $n = 4$, which gives the best result among different settings.

Table 6: Results on ImageNet for domain conversion.

| Methods | Conferences | Cartoon | | Origami | | Toy | | Sculpture | | Average | |
|---|---|---|---|---|---|---|---|---|---|---|---|
| | | R10 | R50 | R10 | R50 | R10 | R50 | R10 | R50 | R10 | R50 |
| Pic2Word [45] | CVPR 2023 | 8.0 | 21.9 | 13.5 | 25.6 | 8.7 | 21.6 | 10.0 | 23.8 | 10.1 | 23.2 |
| Context-I2W [52] | AAAI 2024 | 10.2 | 26.1 | 17.5 | 28.7 | 11.6 | 27.4 | 12.1 | 28.2 | 12.9 | 27.6 |
| Context-I2W* | – | 11.2 | 27.4 | 18.7 | 30.4 | 12.5 | 29.8 | 13.7 | 31.4 | 14.0 | 29.8 |
| Context-I2W(50 %) | AAAI 2024 | 9.0 | 23.0 | 14.3 | 25.6 | 10.7 | 25.0 | 11.0 | 25.5 | 11.3 | 24.8 |
| De-MINDS(50 %) | – | 11.7 | 28.3 | 19.2 | 30.9 | 12.8 | 30.2 | 14.2 | 32.0 | 14.5 | 30.4 |
| **De-MINDS(100 %)** | – | **13.3** | **31.2** | **20.3** | **34.5** | **14.7** | **31.7** | **16.5** | **34.7** | **16.2** | **33.0** |

Table 7: Results on CIRR for object manipulation task.

| Methods | R1 | R5 | R10 | R50 |
|---|---|---|---|---|
| Pic2Word [45] | 23.9 | 51.7 | 65.3 | 87.8 |
| CIReVL [25] | 24.6 | 52.3 | 64.9 | 86.3 |
| LinCIR [20] | 25.0 | 53.3 | 66.7 | – |
| SEARLE-XL [3] | 24.2 | 52.4 | 66.3 | 88.6 |
| SEARLE-XL* | 25.4 | 54.1 | 66.9 | 89.3 |
| Context-I2W [52] | 25.6 | 55.1 | 68.5 | 89.8 |
| Context-I2W* | 26.3 | 55.7 | 69.0 | 90.2 |
| Context-I2W(50%) | 24.8 | 53.6 | 67.1 | 88.9 |
| De-MINDS (50%) | 26.5 | 56.0 | 69.3 | 90.5 |
| **De-MINDS** | **27.3** | **57.0** | **71.3** | **91.6** |

Table 8: Results on COCO for object composition task.

| Methods | R1 | R5 | R10 |
|---|---|---|---|
| Pic2Word [45] | 11.5 | 24.8 | 33.4 |
| Context-I2W [52] | 13.5 | 28.5 | 38.1 |
| Context-I2W* | 14.3 | 29.7 | 40.5 |
| Context-I2W(50%) | 12.1 | 25.6 | 34.4 |
| De-MINDS (50%) | 14.6 | 30.4 | 40.8 |
| **De-MINDS (100%)** | **15.7** | **33.2** | **44.1** |

## A.2 More Effectiveness and Efficiency Analysis

In Table 6 to 9, we present more evidence supporting the efficacy and efficiency of our De-MINDS. With only 50% of the training data, De-MINDS matches and exceeds the performance of the state-of-the-art (SoTA) Context-I2W model by 0.83% to 2.20%. Remarkably, De-MINDS outperforms reported results of the SoTA model by 1.98% to 4.57% under the same 50% training data, underscoring our method's superiority.

Table 9: Results on Fashion-IQ for attribute manipulation.

| Methods | Conferences | Dress | | Shrit | | TopTee | | Average | |
|---|---|---|---|---|---|---|---|---|---|
| | | R10 | R50 | R10 | R50 | R10 | R50 | R10 | R50 |
| Pic2Word [45] | CVPR 2023 | 20.0 | 40.2 | 26.2 | 43.6 | 27.9 | 47.4 | 24.7 | 43.7 |
| CIReVL [25] | ICLR 2024 | 24.6 | 44.8 | 29.5 | 47.4 | 31.4 | 53.7 | 28.6 | 48.6 |
| LinCIR [20] | CVPR 2024 | 20.9 | 42.4 | 29.1 | 46.8 | 28.8 | 50.2 | 26.3 | 46.5 |
| SEARLE-XL [3] | ICCV 2023 | 20.3 | 43.2 | 27.4 | 45.7 | 29.3 | 50.2 | 25.7 | 46.3 |
| SEARLE-XL* | – | 22.7 | 45.0 | 29.4 | 47.9 | 30.2 | 51.4 | 27.4 | 48.1 |
| Context-I2W [52] | AAAI 2024 | 23.1 | 45.3 | 29.7 | 48.6 | 30.6 | 52.9 | 27.8 | 48.9 |
| Context-I2W* | – | 23.9 | 46.9 | 30.4 | 49.7 | 31.1 | 53.8 | 28.5 | 50.1 |
| Context-I2W(50%) | AAAI 2024 | 21.4 | 43.7 | 28.1 | 46.9 | 29.7 | 51.4 | 26.4 | 47.3 |
| De-MINDS (50%) | – | 24.3 | 47.5 | 30.6 | 50.0 | 31.3 | 54.0 | 28.7 | 50.5 |
| **De-MINDS (100%)** | – | **25.2** | **48.7** | **31.0** | **51.2** | **32.9** | **55.7** | **29.7** | **51.9** |

---

**Algorithm 1** Manipulation Intention Understanding's process.

---

**Input**: batch of word embeddings of target descriptions $\tilde{T} = \{t_i\}_{i=1}^m$, where $t_1$ is the global feature $t_{cls}$, $N_{layer}$, the frozen CLIP language encoder $\Psi_T$
**Parameter**: a set of learnable embeddings $X \in \mathbb{R}^{d \times n}$, 8-heads attention layer $Attn$, 3-layers FC layers $f_M$, $gate_\alpha$.
**Output**: target embedding $\hat{t}$

1: Initialize $X \in \mathbb{R}^{d \times n}$, $Attn$, $f_M$ randomly.
2: Let $X_{att}^i = \{t_i\}_{i=2}^m$, $t = 1$
3: **while** $t \leq N_{layer}$ **do**
4:     $X_{att}^{i+1} = X_{att}^i + Attn_t($q=$q$, k=$concat([X_{att}^i, q])$, v=$concat([X_{att}^i, q]))$
5:     $X_{att}^{i+1} = X_{att}^{i+1} + f_{M_t}(X_{att}^{i+1})$
6:     $t = t + 1$
7: **end while**
    $t_* = \Psi_T(X_{output})$
    $\hat{t} = t_{cls} + tanh(gate_\alpha) \cdot t_*$
8: **return** $\hat{t}$

---

## A.3 Broader Impact

We propose a novel image-text dataset augmentation strategy that generates diverse rewrites for any given image-text pair. This approach not only bolsters the performance of vision-language models but also enhances capabilities in textual inversion [44], including text-to-image generation via diffusion models and personalized image retrieval. However, it is crucial to note that MLLMs are trained on extensive web data, which may incorporate factual inaccuracies and hallucinatory content. Consequently, the intention-infused versions of texts could inherit these flaws. We advocate for the implementation of rigorous data filtering methods before these models' deployment in practical settings. Furthermore, while the MLLM-based rewriting strategy demands substantial GPU/TPU computational resources, potentially increasing the carbon footprint.

## A.4 Qualitative Results of intent-CC3M

Figure 9 to 10 we leverage DALL-E [42] to generate images of each caption for qualitative experiment. We compare intent-CC3M with the CC3M dataset and GPT4's rewritten captions. We found that the captions of Intent-CC3M, which contain potential manipulation intentions, provide better visual information compared to the original captions and those rewritten by a large language model. This improvement is due to incorporating diverse visual perspectives (*e.g.,* colors, scenes, and objects) using a multi-model language model, which enhances the training of text-to-image generation tasks. Notably, our pseudo-manipulation descriptions are shorter than the rewritten captions. The results show that pseudo-manipulation descriptions serve as more effective prompts, enabling DALL-E to generate results that are closer to the original images. This demonstrates the high quality of our pseudo-manipulation descriptions.

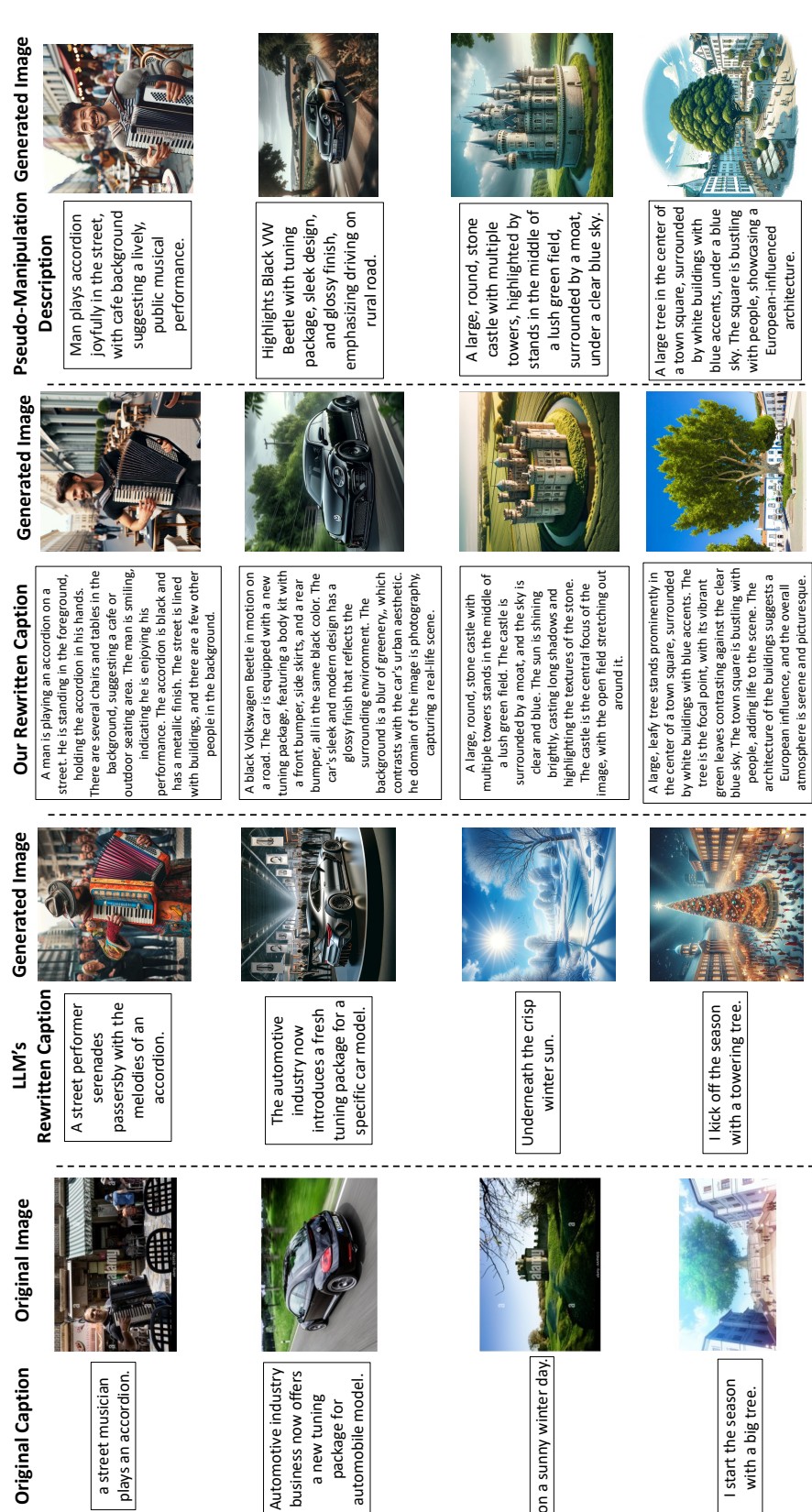

Figure 9: Qualitative results of our intent-CC3M dataset. We leverage DALL-E to generate images of the captions. We compare intent-CC3M with the CC3M dataset and LLM's rewritten captions.

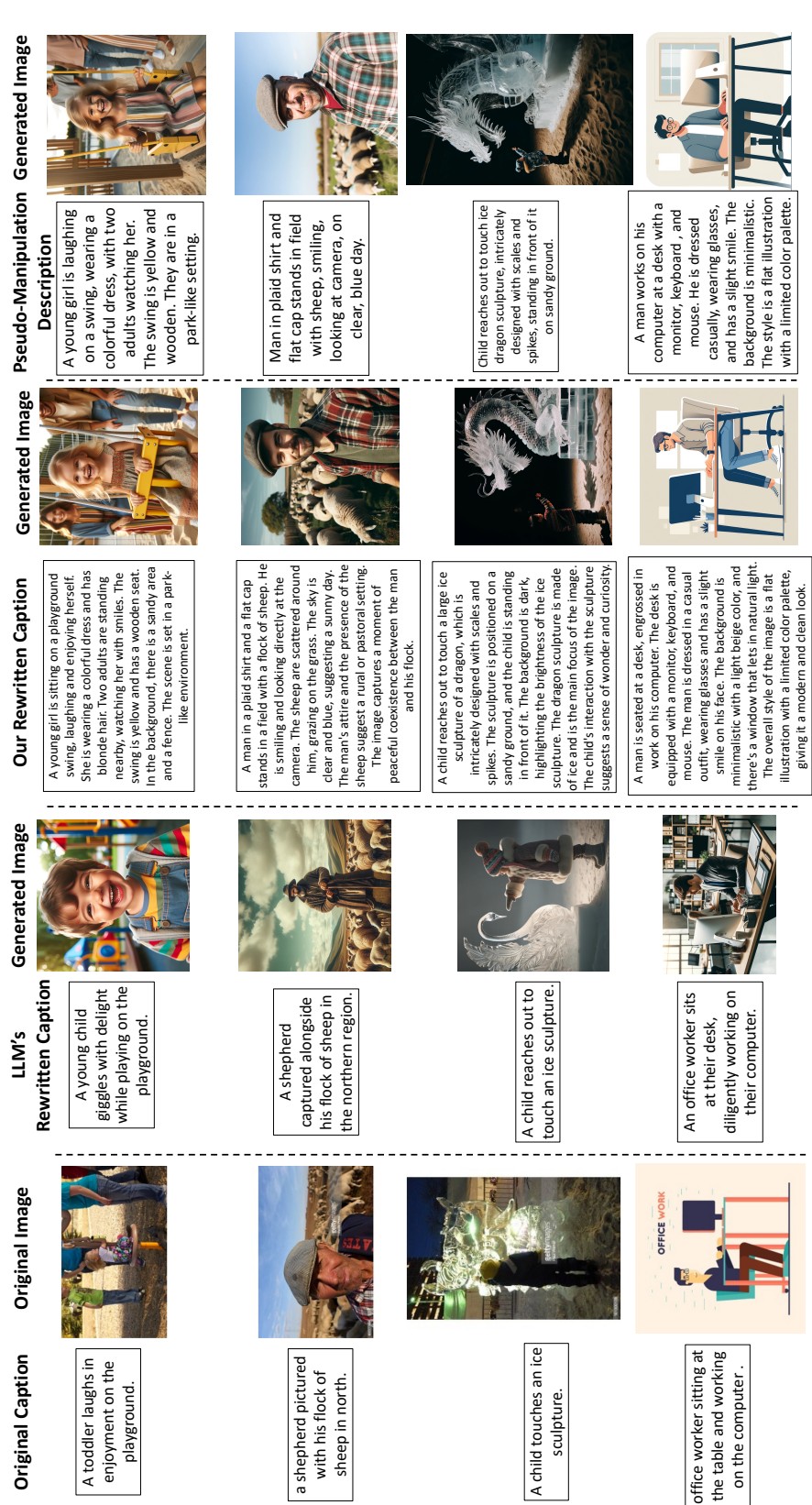

Figure 10: Qualitative results of our intent-CC3M dataset. We leverage DALL-E to generate images of the captions. We compare intent-CC3M with the CC3M dataset and LLM's rewritten captions.

## B    Algorithm of Manipulation Intention Understanding's Process.

Algorithm 1 outlines the pseudo-code for the manipulation intention understanding process. We create a fixed number of learnable embeddings as latent queries to capture intentions that the user aims to modify within manipulation descriptions. These learnable embeddings are then employed in a Transformer to execute cross-attention with the target descriptions word embedding $\{t_i\}_{i=2}^{m}$. The number of output tokens produced by the De-MINDS matches the count of learnable embeddings. To enhance the interaction between learnable embeddings and word embeddings, we concatenate the learnable embeddings with keys and values during the cross-attention process. Each learned query interacts with different intentions, as shown in Figure 2. To achieve a dynamic ratio during the fusion of global and intention embeddings, we utilize a tanh-gating mechanism [23].

Table 10: More ablation study on CIRR and FashionIQ.

|   | Methods | CIRR | | | Fashion-IQ | |
|---|---|---|---|---|---|---|
|   |   | R1 | R5 | R10 | R10 | R50 |
| 1. | 100% original caption | 26.2 | 55.5 | 69.5 | 26.8 | 49.9 |
| 2. | 100% rewritten caption | 25.8 | 55.4 | 69.0 | 26.5 | 49.6 |
| 3. | 100% pseudo-manipulation description | 25.3 | 54.5 | 68.0 | 26.9 | 49.7 |
| 4. | 50% original, 50% rewritten | 26.5 | 55.9 | 70.3 | 27.7 | 50.9 |
| 5. | 50% original, 50% pseudo | 25.5 | 55.2 | 68.6 | 27.0 | 50.1 |
| 6. | 50% rewritten, 50% pseudo | 25.9 | 55.8 | 69.7 | 27.4 | 50.5 |
| 7. | 40% original , 30% rewritten , 30% pseudo | 26.1 | 55.7 | 69.2 | 28.1 | 50.1 |
| 8. | 50% original , 25% rewritten , 25% pseudo | 26.7 | 56.5 | 70.4 | 29.2 | 51.4 |
| 9. | 50% original , 30% rewritten , 20% pseudo | **27.3** | **57.0** | **71.3** | **29.7** | **51.9** |
| 10. | w/o align loss | 20.6 | 45.2 | 57.3 | 23.6 | 42.8 |

## C    Further Ablation Studies on the Training Strategy

Table 10 details additional ablation analyses of the training strategy in De-MINDS. **In model '1-10', we evaluate the necessity of constructs $T$ for pre-training** Our method supports two scenarios in manipulation intention understanding: integrating intention information from lengthy texts and deducing it from concise texts. We evaluated the utility of the original caption $T_{rew}$, the rewritten caption $T_{ori}$, and the pseudo-manipulation description $T_{int}$ in fostering an understanding of manipulation intentions and ensuring training stability. Our experiments led to the optimal ratio of 50% original caption, 30% rewritten caption, and 20% pseudo-manipulation description. Moreover, **in model '9-10', we assess the significance of the alignment loss.** The absence of alignment between the original image embedding and the target embedding in pre-training results in a notable decrease in average performance by 9.54%. This highlights the crucial role of aligning the original image during training, as in CIR, both the reference image and the manipulation intention together create a comprehensive context that defines the target image.

## D    More Details of De-MINDS

### D.1    More Implementation Details For Baseline Models And Mapping Network

Generating one intention caption through LLaVA-1.6-13B [32] for the entire Conceptual Caption dataset [47], which comprises 3M images (CC3M) dataset requires approximately 625 hours on 5 A100 GPUs. By leveraging the capabilities of LLaVA, we ensure that each text sample within the dataset is enriched with diverse and contextually intent-relevant text rewrites, significantly enhancing the dataset's utility for composed image retrieval tasks. For training De-MINDS, we utilize the CC3M and adopt ViT-L/14 CLIP [41] pre-trained on 400M image-text paired data. We employ AdamW [34] with a learning rate of $1 \times 10^{-6}$, weight decay of 0.1, and a linear warmup of 10000 steps. The batch size for contrastive learning is 1024. To improve training stability, we initialize the learnable scalar of tanh-gating to 0 [2]. For training Context-I2W, we only replace the original captions of CC3M with our pseudo-manipulation descriptions. Specifically, we employ AdamW [34] with a learning rate of $1 \times 10^{-5}$, weight decay of 0.1, and a linear warmup of 10000 steps. The batch size for contrastive

learning is 1024. For training SEARLE, we utilize the ImageNet1K [16] test set, which comprises 100K images, and leverage LLaVA to generate intention captions as detailed in Section 3.2. We employ AdamW, with a learning rate of $5 \times 10^{-5}$ and a batch size of 256. All models are trained on 4 NVIDIA A100 (80G) GPUs. Moreover, we conduct ablation studies on CIRR test sets and FashionIQ validation sets. For FashionIQ, we consider the average recall. To ensure reliable results, we report the performance averaged over three trials.

**Mapping network design.** Table 11 summarizes the mapping network $f_\theta$ architecture we employ.

Table 11: Pytorch-style[40] model description of the mapping network $f_\theta$. The output is fed into the CLIP language encoder.

| Layer | Module |
|---|---|
| Output | nn.Linear(512, 768) |
| ReLU2 | nn.ReLU |
| Dropout2 | nn.Dropout(0.1) |
| FC2 | nn.Linear(512, 512) |
| ReLU1 | nn.ReLU |
| Dropout1 | nn.Dropout(0.1) |
| FC1 | nn.Linear(512, 512) |

## D.2 More Evaluation Datasets Details of Query and Candidate Images.

We evaluate our model on four ZS-CIR datasets, *i.e.,* COCO [31] for object composition, ImageNet [16, 21] for domain conversion, CIRR [33] for object/scene manipulation, and Fashion-IQ [57] for attribute manipulation. All the dataset settings and evaluation metrics (Recall@K) follow the recent works [45, 52] for a fair comparison. The evaluation datasets are preprocessed, as explained in the main paper, we describe the details of the dataset, *i.e.,* number of query images and candidate images used for evaluation.

Table 12: The number of images used for evaluation in each dataset.

| Dataset | Query images | Candidate images |
|---|---|---|
| ImageNet | 10,000 | 16,983 |
| COCO | 4,766 | 4,766 |
| CIRR (test) | 4,148 | 2,315 |
| Fashion (Dress) | 2,017 | 3,817 |
| Fashion (Shirt) | 2,038 | 6,346 |
| Fashion (TopTee) | 1,961 | 5,373 |

## D.3 More Inference Details of Prompts for Different Evaluate Tasks

**(1) Domain conversion**. This setup evaluates the ability to compose real images and domain information to retrieve corresponding domain-specific images. We utilize ImageNet [16] and ImageNet-R [21], which comprises 200 classes with diverse domains and has domain annotations. Following Pic2Word, we pick cartoon, origami, toy, and sculpture as the evaluation target to avoid noise in the annotations. With this selection, we have 16,983 images as candidates. In the evaluation, given the real image from ImageNet and target domain names, we compose the query following the procedure in (a) in the Inference section. *e.g.,* a `cartoon of [*]`.

**(2) Object composition**. We evaluate the validation split (5000 images) of COCO [31], which dataset contains images with corresponding lists of object classes and instance mask of query images. Following Pic2Word, we randomly crop one object and mask its background using its instance mask to create a query for each image. The list of object classes is used as text specification. Given the reference image and class list, we compose a query by following (b) in the Inference section. *e.g.,* a `photo of [*], [cat] and [dog]`.

**(3) Object/scene manipulation by text description**. In this setup, a reference image is provided alongside a text description containing instructions for manipulating either an object or the background

scene depicted in the reference image. This composition of the reference image and text description enables the retrieval of manipulated images. We evaluate the test split of CIRR [33] using the standard evaluation protocol following previous works [45, 3, 52], and query texts are composed following the procedure in (c) of the Inference section.

**(4) Attribute manipulation**. We employ Fashion-IQ [57], which includes various modification texts related to image attributes. These attribute manipulations are given as a sentence. As with CIRR, we adopt the standard evaluation protocol and create query texts following the procedure provided in (c) of the Inference section. In evaluation, we employ the validation set, following previous works [4, 45, 3, 52].

# E   Extended Related Works

**Mapping Image as One Word.** Several methods [30, 59] represent image regions as word tokens via VLP models, which rely on object detector efficacy. However, ZR-CIR tasks extend the alignment ability beyond objects to scenes, styles, attributes, *ect*. Our method addresses this issue by employing pseudo triplet data, which maps a pseudo reference image to a pseudo word token and combines it with the caption to align with the target image. PALAVRA [14] proposes personalized image retrieval via cycle contrastive loss, requiring class-wise and caption annotations. In contrast, our model facilitates fine-grained image-to-word mapping without additional annotations. Other approaches [26, 36, 62, 50] utilize a single word token to represent multiple images of the same object for text-to-image generation. Our model obviates the need for costly image-supervised training.

**Knowledge Distillation.** Knowledge distillation is a machine learning technique wherein a simpler model, known as the student, learns to mimic the behavior of a more complex model, known as the teacher, by learning from its predictions [22]. This approach has demonstrated efficacy across various computer vision tasks, including image classification [22, 43, 5], object detection [9, 8], and text-to-image synthesis [35, 46], resulting in improved model compression, computational efficiency, and accuracy. In our study, we employ knowledge distillation to transfer knowledge from a computationally expensive optimization method (teacher) to a more lightweight neural network (student). Specifically, we train a manipulation intention understanding network to replicate the reasoning ability of an MLLM using a distillation loss. Alternatively, our lightweight network can be interpreted as a surrogate model of the more resource-intensive technique.

