# OpenReview forum: "Manipulation Intention Understanding for Accurate Zero-Shot Composed Image Retrieval"
_NeurIPS.cc/2024/Conference — Submitted to NeurIPS 2024_

### Official Review · Reviewer_6wVo · 2024-07-02

**Soundness:** 2
**Presentation:** 2
**Contribution:** 2
**Rating:** 4
**Confidence:** 4

**Summary:**

This paper focuses on Zero-Shot Composed Image Retrieval (ZS-CIR), which requires retrieving an image matching a reference image while incorporating specified textual modifications. The authors argue that a key challenge in ZS-CIR is training models on limited intention-relevant datasets to understand human intention implicitly expressed in textual modifications for accurately retrieving target images. Therefore, they introduce an image-text dataset incorporated with pseudo-manipulation intentions to enhance the training of ZS-CIR models in understanding human manipulation intents, based on LLaVA. They also propose to use Q-former to compress the features generated by CLIP for retrieval. The experimental results show the improvements of the proposed method.

**Strengths:**

1. The authors propose a large-scale pretraining dataset for ZS-CIR.
2. The experimental results show the improvements of the proposed method.

**Weaknesses:**

1. The concept of "intention" discussed throughout the whole paper is unclear. Based on Figure 1, the authors haven't explained what's "intention" to the MLLM. Moreover, the pseudo-manipulation description is the same as the rewritten caption in semantics. I can't find any "intention" added into this pseudo-manipulation description. The key novelty of considering human "intention" is farfetched.
2. The proposed model lacks novelty. In the model architecture, the authors just add a Q-Former [1] after the CLIP encoder, which is prevalent in existing research based on CLIP-like models. And the authors even do not cite any relevant work.
3. Existing work on CLIP-based ZS-CIR generally compares the experimental results with different CLIP variants, and different methods may be superior with different CLIP variants. The authors only experimented with one CLIP variant, which is insufficient.
4. In Figure 4, the compared method also accurately captures the "intention" in the modification text, which cannot show the superiority of the proposed method in capturing "intention".

References:

[1] Li, J., Li, D., Savarese, S., & Hoi, S. (2023, July). Blip-2: Bootstrapping language-image pre-training with frozen image encoders and large language models. In International conference on machine learning (pp. 19730-19742). PMLR.

**Questions:**

What's human manipulation intention discussed in this paper? Why do the original captions have little intention? Why do the pseudo-manipulation descriptions contain more intention?

**Limitations:**

The authors addressed the limitations.

---

> ### Author Rebuttal · Authors · 2024-08-05
>
> **W1,Q1. Detailed Explanation of "Intention" in Our Work**
>
> Thank you for your question! In our manuscript (lines 51-57), we define "intention" as the implicit, latent intent within manipulation descriptions. For better clarity, we visualize comparisons between the original captions, rewritten captions, and pseudo-manipulation descriptions in Figure 2 of our Author Rebuttal PDF (samples from Appendix A.4), illustrating two key forms of intention,
>
> - **Intentions Embedded in Abstract Text**: As shown in Line 1 of Figure 2,caption like '`a sunny winter day`' implies hidden intention such as '`sun shining brightly`' and `'sky is clear`'. Similarly, in Line 3, '`working on computer`' implies intention like '`equipped with a monitor, keyboard, and mouse`'. These embedded intentions, while obvious to humans, is challenging for models due to a lack of intention-specific training data. We address this by leveraging MLLMs' reasoning capabilities to explicitly articulate these hidden intentions in rewritten captions, training CIR models to understand user intentions in abstract text.
> - **Intentions Hidden in Redundant Text**: As illustrated in Line 2 of Figure 2, the rewritten captions include redundant intention-irrelevant visual details (e.g., '`coexistence between the man and his flock`'). These details pose challenges for models to understand the manipulation intention from text. By employing MLLMs, we effectively filter out these intention-irrelevant visual details, resulting in pseudo-manipulation descriptions to train the model to focus on the essential manipulation intention.
>
> We appreciate your feedback and will add a detailed explanation in the revised manuscript.
>
> ---
>
> **W2. Technology Contribution of Our De-MINDS Framework**
>
> We appreciate your valuable comment! The key novelty of our technology contribution is the entire De-MINDS framework rather than the Q-Former component. Our framework addresses the challenge of accurately understanding user intention in manipulation descriptions. While existing Q-former-based models focus on filtering out task-irrelevant visual information. We clarify the key difference from two aspects:
>
> - **Existing research falls short in interpreting user intention due to the lack of intention-based training data.** Our ablation studies (model #2 in the manuscript) show that the model without pseudo-manipulation descriptions results in considerable performance decline, proving the collaborative effectiveness of the whole De-MINDS framework beyond the Q-former component.
>
> - **The Q-former-based component plays a critical role but is not the sole component in our framework**. After obtaining pseudo-manipulation descriptions, we aim to develop a module to capture intention-relevant information from abstract or redundant text. We aim to train a Q-former-based method to distill the intention understanding ability of MLLMs. Our ablation studies (i.e., model “7-8”) in the manuscript,  further illustrate the contribution of other modules, proving that without reasoning distillation leads to a significant performance decline.
>
> Thus, our technical contribution is the entire De-MINDS framework to understand manipulation intentions in user descriptions for CIR . Also, we want to clarify that we have cited Q-former+CLIP-like models in the "Related Works" Section [1, 28, 38, 37, 52] in our manuscript.
>
> ---
>
> **W3. Insufficient Experiment on One CLIP Backbones**
>
> Thank you for your suggestion! We have conducted experiments with the ViT-G and the ViT-H. Please refer to the response to the common concerns for detailed results and analysis.  We will add them in the revised manuscript.
>
> ---
>
> **W4. The Superiority of the Proposed Method in Capturing Intention**
>
> Thank you for your insightful feedback! CIR task requires a model to guarantee two aspects of abilities:
>
> - preserving details from the manipulation language, and
> - maintaining fidelity to the visual content of the reference image.
>
> As shown in Figure 4, although existing models transfer the image domain as required correctly, they struggle to maintain the consistency of the main visual content in the reference image, and thus, do not fully capture the user's intention. For instance, the last row in Figure 4 shows a user's intent to transform a seal image into a cartoon domain. Existing models gain pseudo-tokens with redundant information (e.g., rocks and water),  fail to correctly focus on the key element of  '`a cartoon image of seal`'. In contrast, our De-MINDS filters out these irrelevant visual details, leading to more accurate CIR results (as detailed in lines 286-291 of our manuscript).
>
> ---
>
> **Q2. Why Do the Original Captions Have Little Intention?**
>
> Thank you for your insightful question! As illustrated in Figure 2 of our Author Rebuttal PDF, the original captions in image-caption datasets like CC3M are often brief and abstract, lacking intention-relevant information crucial for CIR tasks. This absence appears in two main forms:
>
> - **Absence of Visual Detail:** For instance, Line 1 in Figure 2 describes the general scene of '`a sunny winter day`' but omits crucial details about significant objects within that scene  (e.g.,  '`castle`').
> - **Absence of Style or Attribute Details:** For example, Line 3 in Figure 2 mentions the main subjects (e.g., man and computer) but fails to refer to the cartoon style of the image, which is essential for domain-specific manipulations.
>
> The information absence in dataset captions significantly hinders the training of models to understand and manipulate images based on detailed user intention (e.g., objects, scenes, colors, and styles).
>
> ---
>
> **Q3. Why do the pseudo-manipulation descriptions contain more intention?**
>
> Thank you for your question! It results from our Chain-of-Thought prompting strategy to filter out irrelevant visual details, refining the descriptions to include more intentions. For more details please refer to **Intentions Hidden in Redundant Text** of **W1,Q1**.

---

> > ### Comment · Area_Chair_SMz3 · 2024-08-12
> > **Official Comment by Area Chair SMz3**
> >
> > Dear reviewer 6wVo, as the NeurIPS AC, I would like to remind you to check and read the rebuttal provided by the authors. After that, please respond or acknowledge authors ' rebuttal and update your rating if necessary. Many thanks.

---

### Official Review · Reviewer_KSmi · 2024-07-09

**Soundness:** 3
**Presentation:** 2
**Contribution:** 3
**Rating:** 6
**Confidence:** 4

**Summary:**

This paper introduces De-MINDS, a novel framework for Zero-Shot Composed Image Retrieval (ZS-CIR) that aims to bridge the gap between pre-training and retrieval by incorporating intention-based pseudo-manipulation descriptions. The authors propose intent-CC3M, a dataset featuring these descriptions generated through chain-of-thought prompting by a Multi-modal Large Language Model (MLLM). They also introduce a manipulation intention understanding network that uses learnable queries to enhance the model's ability to understand user intentions from manipulation descriptions. The paper demonstrates significant performance improvements across four ZS-CIR tasks compared to state-of-the-art models.

**Strengths:**

- The introduction of intent-CC3M as a dataset for training mapping networks to align intention-relevant visual information is innovative and potentially impactful.
- The proposed De-MINDS framework shows significant performance improvements over state-of-the-art models across multiple ZS-CIR tasks.
- The approach addresses the challenge of understanding manipulation intentions in user descriptions, which is crucial for accurate image retrieval.
- The ablation studies provide insights into the contributions of different components of the proposed method.

**Weaknesses:**

Major Weaknesses:

1. Experimental Gaps:
   - The paper lacks experimental evidence to support the claim that caption redundancy leads to inaccurate retrieval, as mentioned in the introduction.
   - There's no evaluation of the method's performance with longer text encoders like LongCLIP, which could potentially address some of the stated limitations of CLIP.
   - The comparison with a baseline (other than CIRR and Fashion-IQ) using only f_theta (trained on Intent-CC3M) without De-MINDS (ablation model '4') is missing, which would provide a fairer comparison.

2. Methodological Concerns:
   - The justification for using CC3M as the base dataset for creating intent-CC3M is not clearly explained.
   - There's no exploration of De-MINDS' performance when prompt options are mismatched with their intended tasks or in scenarios where the task is not known in advance.

3. Incomplete Ablation Studies:
   - The ablation study for the T sampling ratios (50%, 30%, 20%) is missing, and there's no explanation why concatenation of them wasn't considered as an alternative.
   - The ablation study lacks an exploration of the impact of the number of learnable queries, despite its apparent significance.

Minor Weaknesses:

1. Presentation Issues:
   - The prompt types (a), (b), and (c) are not clearly explained in the context they are introduced, requiring readers to refer back to previous sections.
   - There are inconsistencies between the notation in the text and figures (e.g., X vs q in Figure 2).

2. Comparative Analysis:
   - The paper doesn't include evaluations on CIRCO and GeneCIS datasets, which were used in baseline studies.

3. Clarity:
   - More details are needed on certain aspects, such as the "cos distill" mentioned in ablation model '9'.

**Questions:**

- How does the performance of De-MINDS compare to existing methods when intention is injected as "raw text" into their frameworks?
- Can you provide more details on the "cos distill" mentioned in ablation model '9'?
- Have you considered conducting an ablation study on the number of learnable queries used in De-MINDS?
- How does De-MINDS perform when the task is not known in advance? Is there a possibility of developing a "general intention" version of De-MINDS?
- Can you elaborate on why CC3M was chosen as the base dataset for creating intent-CC3M?

**Limitations:**

The authors acknowledge the computational intensity of generating pseudo-manipulation descriptions using MLLMs and the potential introduction of irrelevant details in these descriptions. However, they could further discuss the implications of these limitations on the practical applicability of their method in real-world scenarios.

---

> ### Author Rebuttal · Authors · 2024-08-05
>
> **W1.  Lacks evidence to support the claim that caption redundancy leads to inaccurate retrieval**
>
> We appreciate your valuable feedback!  As demonstrated in Figure 1 of our PDF in Author Rebuttal, caption redundancy presents significant challenges of the SoTA model (i.e., Context-I2W) from two perspectives:
>
> - **Visual Perspective:** It makes the model struggle to filter out manipulation text-relevant information.
> - **Textual Perspective:** It makes the model difficult to accurately attend to textual information related to the user intention described in the manipulation text (e.g., object, scenes, color, and style).
>
> To address these issues, as illustrated in Figure 7, our De-MINDS distils the reasoning ability of MLLMs, enhances the intention understanding ability of the CLIP. This mitigates the challenge, improving CIR models' ability to understanding manipulation intentions in user descriptions. We will add these results into the revised manuscript.
>
> ---
>
> **W2.  No evaluation of the method's performance with longer text encoders**
>
> We have conducted additional experiments with the LongCLIP model. Please refer to the response to the common concerns for detailed experimental results and analysis. We appreciate your valuable feedback and will add these results in the revised manuscript.
>
> ---
>
> **W3. Need Ablation Studies on COCO and ImageNet**
>
> Thank you for your insightful feedback! Fashion-IQ and CIRR are widely compared datasets in the existing works of ZS- CIR. Therefore, we follow prior works [1,2,3] to conduct ablation studies on these datasets for fair comparison. As suggested by the reviewer, in the Tables below, we have further conducted experiments of the ablation model #4 on COCO and ImageNet. The results are consistent with others. We will add them to the revised manuscript.
>
> | Method         | R@1  | R@5  | R@10 |
> | :------------- | :--: | :--: | :--: |
> | w/o De-MINDS   | 12.1 | 26.3 | 35.5 |
> | general prompt | 15.3 | 32.8 | 43.6 |
> | **De-MINDS**   | 15.7 | 33.2 | 44.1 |
>
> | Method         |  Cartoon  |  Origami  |    Toy    | Sculpture |  Average  |
> | -------------- | :-------: | :-------: | :-------: | :-------: | :-------: |
> |                | R@10 R@50 | R@10 R@50 | R@10 R@50 | R@10 R@50 | R@10 R@50 |
> | w/o De-MINDS   | 9.2 23.1  | 14.6 26.8 | 10.1 24.2 | 11.3 25.2 | 10.1 23.2 |
> | general prompt | 12.7 30.6 | 19.8 33.9 | 14.2 31.1 | 16.0 34.2 | 15.7 32.5 |
> | **De-MINDS**   | 13.3 31.2 | 20.3 34.5 | 14.7 31.7 | 16.5 34.7 | 16.2 33.0 |
>
> ---
>
> **W4,Q5.  Justification for Using CC3M as the Base Dataset**
>
> Thank you for your insightful comment! CC3M is widely compared in existing ZS-CR methods [1, 2, 3] due to its diverse visual details (e.g., object, scenes, color, and style). This diversity makes De-MINDS learn various kinds of manipulation intentions. Additionally, we used ImageNet to generate pseudo-manipulation descriptions for training SEARLE-XL*, as detailed in Appendix D.1 and Section 4.1, achieving consistent performance improvement.
>
> ---
>
> **W5,Q4. De-MINDS' Performance On General Prompt**
>
> Thank you for your insightful feedback! De-MINDS is trained with a task-agnostic, general prompt (i.e., `a photo of S*, [caption]`), making it effective in supporting "general intention". For inference, as mentioned in our manuscript, to focus on the study of manipulation intention understanding for ZS-CIR, we use the prompts the same as  recent works [1,2] for a fair comparison. Moreover, we add experiments on COCO and ImageNet using a general prompt  (i.e., `a photo of [*], [sentence]`), as shown in the above Tables, which obtains consistent results as others, proving De-MINDS's adaptability to unknown tasks. Thank you for the reviewer’s suggestion, we will further study the influence of prompt options in our future work.
>
> ---
>
> **W6,W7,Q2.  Ablation Studies of Hyperparameters**
>
> Thank you for your suggestion! The ablation of hyperparameters has indeed been considered. Detailed ablation on T sampling ratios and the number of learnable queries are in Appendix C and Appendix A.1 of our manuscript, respectively. We will include these results in the main part of the revised manuscript.
>
> ---
>
> **W8,W9.  Presentation Issues**
>
> We appreciate you pointing out these issues! We will add clearer explanations of the prompt types and clarify inconsistencies in notation in Figure 2 in the revised manuscript.
>
> ---
>
> **W10.  Evaluations on different datasets**
>
> Thank you for your insightful feedback! Since Fashion-IQ and CIRR are the most commonly compared datasets in the CIR tasks in the existing works [1,2,3], we chose them to validate our method across different aspects (e.g., domain, objects) using ImageNet and COCO , which also utilized baseline models [1, 2]. Due to the time limitation of the rebuttal phase, we will add results on CIRCO and GeneCIS in our revised manuscript.
>
> ---
>
> **W11,Q2. Details of the "cos distill"**
>
> Thank you for pointing out this issue!   "cos distill" means using cosine distillation loss, denoted as `L_cos = 1 - cos(s, t)`, for reasoning distillation, instead of contrastive loss. We will add these explanation in the revised manuscript.
>
> ---
>
> **Q1. Performance of Existing Methods With Intention Injected as "raw text"**
>
> Thank you for your insightful question! We have considered such a problem and conducted experiments in our work. In our manuscript, Tables 1~4 compare the performance of the existing methods with their performance when trained on our pseudo-manipulation descriptions as raw text (i.e., `SEARLE-XL*` and `Context-I2W*` ) . The results show consistent performance improvement when the intention is injected as “raw text” into their frameworks.
>
> **References**
>
> [1] Pic2word: Mapping pictures to words for zero-shot composed image retrieval, CVPR 2023.
>
> [2] Context-I2W: Mapping Images to Context-dependent Words for Accurate Zero-Shot Composed Image Retrieval,  AAAI 2024.
>
> [3] Language-only training of zero-shot composed image retrieval, CVPR 2024.

---

> > ### Comment · Area_Chair_SMz3 · 2024-08-12
> > **Official Comment by Area Chair**
> >
> > Dear reviewer KSmi, as the NeurIPS AC, I would like to remind you to check and read the rebuttal provided by the authors. After that, please respond or acknowledge authors ' rebuttal and update your rating if necessary. Many thanks.

---

> > ### Comment · Reviewer_KSmi · 2024-08-13
> >
> > Thank you for your detailed response to my review. Your additional experiments and explanations have addressed many of the concerns raised, particularly regarding the evidence for caption redundancy issues, the performance with longer text encoders, and the justification for using CC3M as the base dataset. The ablation studies on COCO and ImageNet, as well as the clarification on De-MINDS' performance with general prompts, provide valuable insights into the model's capabilities and generalizability. I appreciate the thoroughness of your responses and maintain my rating as-is.

---

> > ### Author Response · Authors · 2024-08-13
> >
> > Dear Reviewer KSmi,
> >
> > Thank you once again for your time and thoughtful feedback! We're pleased to hear that our responses have addressed many of your concerns. We sincerely appreciate your efforts in reviewing our paper and your insightful comments. Your support and constructive feedback have been invaluable.
> >
> > Best regards,﻿
> >
> > Authors of Submission 1089

---

### Official Review · Reviewer_CKWd · 2024-07-10

**Soundness:** 3
**Presentation:** 3
**Contribution:** 3
**Rating:** 5
**Confidence:** 5

**Summary:**

This paper introduces an image-text dataset (intent-CC3M) for Zero-Shot Composed Image Retrieval (ZS-CIR) models to make better understanding of human manipulation intentions. Specifically, captions are re-written with LLaVA model to provide more details, and additional manipulation reasoning prompt is applied to make pseudo-manipulation description. With this dataset, the paper proposes De-MINDS framework (unDErstanding of Manipulation INtention from target Description before Searching), which utilizes pseudo-manipulation descriptions. The model training involves reasoning distillation and cross-modal alignment. The method shows state-of-the-art performance with ViT-L backbone comparisons.

**Strengths:**

This paper proposes to leverage LLaVA model to elaborate the image caption and further utilize LLaVA's reasoning capability to build a pseudo manipulation. The proposal is intuitive and clear, and the presentation of this paper is also clear. Extensive results including various ablations and qualitative results demonstrate the proposed method.

**Weaknesses:**

The proposed method of utilizing LLM, referred to as MLLM, is not entirely novel, as it has been previously addressed in works [1, 2] (please also refer to [1]). Furthermore, the evaluation of the proposed method is limited to the ViT-L backbone, which raises concerns about its effectiveness with other, more robust backbones (such as ViT-G).

[1] Jang, et al. Visual Delta Generator with Large Multi-modal Models for Semi-supervised Composed Image Retrieval, CVPR2024
[2] Karthik, et al, Vision-by-Language for Training-Free Compositional Image Retrieval, ICLR2024

**Questions:**

Since the proposed method utilize LLaVA model's generated result, would it be any investigation on hallucination?

**Limitations:**

The paper handles possible limitations properly.

---

> ### Author Rebuttal · Authors · 2024-08-05
>
> **W1. Distinctiveness of the De-MINDS Framework Compared to Other LLM-based CIR Methods**
>
> We appreciate the reviewer's insights about the novelty of our De-MINDS compared with existing LLM-based CIR approaches [1, 2]. Although De-MINDS employs Large Language Models (LLMs), the motivation of our work is fundamentally different from [1, 2] as follows,
>
> - [1] discusses the utilization of LLMs to "transfer knowledge from Large Language Models (LLMs) and connect it with semi-supervised Composed Image Retrieval".
> - [2] focuses on methods that "match or outperform existing training-based methods on four CIR benchmarks while only relying on off-the-shelf available pre-trained models"
>
> - Our De-MINDS aims to capture the user intention in manipulation descriptions by specific model design based on  the reasoning ability of MLLMs.
>
> Meanwhile, De-MINDS is also different from [1,2] in the aspect of technical novelty. Unlike the approaches in [1] and [2] that either fine-tune or directly apply LLMs during the inference stage of the CIR tasks, *our De-MINDS does not employ MLLMs directly during the inference stage to improve the inference accuracy and efficiency*. Instead,
>
> - We utilize a novel Chain-of-Thought prompting strategy with MLLMs to initially generate intention-based pseudo-manipulation descriptions. These descriptions are then used to train a lightweight manipulation intention understanding network to distill MLLM’s capabilities of interpreting user intentions for accurate CIR.
>
> - This strategy achieves a significant improvement of 2.05%～4.78% on complex intention datasets (i.e., Fashion-IQ and CIRR) compared to the LLM-based ZS-CIR model (e.g., CIReVL [2]) as shown in Table 1 to Table 2 below (results are collected from our manuscript),
>
>   ###### Table 1: Results on the Fashion-IQ Dataset for Attribute Manipulation: De-MINDS vs. LLM-based ZS-CIR Method.
>
>   | Method   |       Dress       |      Shirt      |     Toptee      |     Average     |
>   | -------- | :---------------: | :-------------: | :-------------: | :-------------: |
>   |          |   R@10    R@50    |  R@10    R@50   |  R@10    R@50   |  R@10    R@50   |
>   | CIReVL   |   24.6     44.8   |   29.5   47.4   |   31.4   53.7   |   28.6   48.6   |
>   | De-MINDS | **25.2     48.7** | **31.0   51.2** | **32.9   55.7** | **29.7   51.9** |
>
>   ###### Table 2: Results on the CIRR Dataset for object Manipulation: De-MINDS vs. LLM-based ZS-CIR Method.
>
>   | Method   |   R@1    |   R@5    |   R@10   |   R@50   |
>   | :------- | :------: | :------: | :------: | :------: |
>   | CIReVL   |   24.6   |   52.3   |   64.9   |   86.3   |
>   | De-MINDS | **27.3** | **57.0** | **71.3** | **91.6** |
>
>
> - Moreover, our De-MINDS achieves real-time inference speed. Our inference time (0.017s) is ×58 faster than CIReVL (∼ 1s), which uses LLM for inference, underscoring the potential ability of De-MINDS for real-world applications.
>
> Thank you for your valuable feedback! We will include the detailed analysis in the "Related Works" section of our revised manuscript, ensuring citation of reference [1].
>
> ---
>
> **W2. Effectiveness of De-MINDS with More Robust Backbones**
>
> Thank you for your constructive feedback! We have conducted additional experiments with the ViT-G and the ViT-H models. Please refer to the response to the common concerns for detailed experimental results and analysis. We appreciate your feedback and will add these results into the revised manuscript.
>
> ---
>
> **Q1. Investigating Hallucination in MLLM-Generated Results**
>
> Thank you for pointing out the important issue of model hallucinations associated with our use of MLLMs. We have considered the potential issue of hallucinations and have specific design in our model to mitigate this risk by two strategies:
>
> - **Careful prompt design.**  As demonstrated in the code (`cc_rewrite_multi.py`) in our supplementary materials, we crafted prompts to "Minimize aesthetic descriptions as much as possible," which significantly reduces hallucinations, as proved by the findings in [3].
> - **Chain-of-Thought prompting strategy.** We simplify complex tasks into manageable steps, using the MLLM multiple times for each image-caption pair, thus decreasing hallucination likelihood. Our manuscript illustrates this in Figure 2 with the "*Original Caption Rewriting Process*," which uses MLLM's reasoning to extract potential human manipulation intentions from different visual perspectives, integrating these into the rewritten captions. This is followed by the "*Pseudo-Manipulation Description Reasoning Process*," which involves re-invoking the MLLM with these captions and the original image to generate intention-based Pseudo-Manipulation Descriptions. Figures 9-10 (in our manuscript) showcase how this strategy effectively ensures the accuracy of MLLM outputs, minimizing hallucination risks.
>
> The above two strategies for decreasing the influence of hallucinations are effective not only for our De-MINDS, but also for existing approaches. In Tables 1~4 in our manuscript, we show the results of existing SoTA approaches, i.e., `SEARLE-XL*` and `Context-I2W*`, using our pseudo-manipulation descriptions for training. It shows consistent performance improvement over existing ZS-CIR methods. This proves the generalizability and effectiveness of our method for hallucinations associated with the use of MLLMs .
>
> **References**
>
> [1] Jang, et al. Visual Delta Generator with Large Multi-modal Models for Semi-supervised Composed Image Retrieval, CVPR2024
>
> [2] Karthik, et al, Vision-by-Language for Training-Free Compositional Image Retrieval, ICLR2024
>
> [3] Chen, Lin, et al. Sharegpt4v: Improving large multi-modal models with better captions, ECCV 2024.

---

> ### Comment · Reviewer_CKWd · 2024-08-07
>
> Thanks for your concrete response.
>
> For W1, most of my concerns are resolved. One thing to note that is [1] does not utilize additional inference cost of MLLM so it will be fast, and it would be more stronger paper if you include retrieval speed of De-MINDS, comparing with others. Also, is there any specific reason for naming "attribute manipulation" and "object manipulation" for standard FashionIQ and CIRR benchmarks? If it is not different with standard benchmark, it would be better to name it simply FashionIQ and CIRR to make it clear.
>
> For Q1, I also agree author carefully tries to handle the hallucination well.

---

> ### Author Response · Authors · 2024-08-08
>
> Thank you for your time  and insightful suggestions! We appreciate your suggestion and respond to each point as follows.
>
> ---
>
> **Suggestion 1: Inclusion of Retrieval Speed of De-MINDS Compared to Other Models**
>
> We appreciate your valuable suggestion! We have indeed considered the retrieval speed of De-MINDS and compared it with other ZS-CIR methods in "**Effectiveness and Efficiency Analysis**" (Section 4.3) of our manuscript:
>
> - Our inference time (0.017s) is ×58 faster than CIReVL (∼ 1s), which uses LLM for inference, and only 0.005s slower than Pic2Word, which employs a simple 3-layer MLP for mapping.
>
> [1] is an inspiring work that does not utilize additional cost of MLLM for fast inference. As you suggested, we will compare the inference time of [1] with De-MINDS in Section 4.3 in our revised manuscript
>
> ---
>
> **Suggestion 2:  It would be better to name "attribute manipulation" and "object manipulation" simply Fashion-IQ and CIRR to make it clear.**
>
> Thank you for your suggestion! We adopt specific terms "attribute manipulation" and "object manipulation" following the expression of existing works [1,2], which highlight the unique aspects of human manipulation descriptions inherent in each dataset,
>
> - **Fashion-IQ** (e.g., Figure 3) is employed to assess the manipulation of fashion image attributes, typically involving changes to clothing attributes.
> - **CIRR** (e.g., Figure 5) is employed to evaluate manipulations involving objects or background scenes in real-life images.
>
> We agree with the reviewer that clarity and consistency in terminology is quite important. We will provide additional explanations in the "**Dataset**" section in our manuscript and will adopt more straightforward terms “Fashion-IQ” and “CIRR” in the "**Experiment**" section to improve the clarity.
>
> **References**
>
> [1] Jang, et al. Visual Delta Generator with Large Multi-modal Models for Semi-supervised Composed Image Retrieval, CVPR2024.
>
> [2] Saito, Kuniaki, et al. Pic2word: Mapping pictures to words for zero-shot composed image retrieval, CVPR 2023.
>
> [3] Tang, Yuanmin, et al. Context-I2W: Mapping Images to Context-dependent Words for Accurate Zero-Shot Composed Image Retrieval, AAAI 2024.

---

> ### Author Response · Authors · 2024-08-13
>
> Dear Reviewer CKWd,
>
> Thank you once again for your time and insightful suggestions! We would like to confirm whether our responses have addressed your concerns. If our clarifications are satisfactory, we hope our response may merit raising your rating.
>
> Best regards,﻿
>
> Authors of Submission 1089

---

> ### Author Response · Authors · 2024-08-14
>
> Dear Reviewer CKWd,
>
> Thank you once again for your time and thoughtful feedback! We noticed that you raised your score of our contribution from "fair" to "good." This adjustment is incredibly meaningful to us and has a significant impact on our paper. We are truly grateful for your careful consideration and the positive reflection of our work.
>
> Your support and constructive insights have greatly contributed to improving our submission. We sincerely appreciate your efforts in reviewing our paper.
>
> Best regards,
>
> Authors of Submission 1089

---

### Author Rebuttal · Authors · 2024-08-05

Dear Reviewers,

We sincerely thank you for your insightful feedback! We are encouraged by the positive comments such as " The proposal is intuitive and clear" (Reviewer CKWd), "The introduced intent-CC3M dataset is innovative and potentially impactful" (Reviewer KSmi) , "the presentation is also clear." (Reviewer CKWd), "The proposed De-MINDS framework shows significant performance improvements over SoTA models across multiple ZS-CIR tasks."(Reviewer KSmi, CKWd, 6wVo) , "The approach addresses the challenge of understanding manipulation intentions in user descriptions, which is crucial for accurate image retrieval." (Reviewer KSmi), "Extensive results including various ablations and qualitative results demonstrate the proposed method." (Reviewer KSmi, CKWd).

Response to the common concerns raised by the reviewers.

**Concerns on De-MINDS Effectiveness Across Robust Backbones and with Longer Text Encoder**

For a fair comparison with previous studies [1, 2], we follow their settings to utilize ViT-L as the backbone of our model. Following the suggestion of Reviewers **CKWd** and **6wVo**, we have conducted the experiments with ViT-H and ViT-G backbones, utilizing checkpoints from OpenCLIP [3]. Moreover, in response to Reviewer **KSmi**'s concern, we have conducted experiments with the LongCLIP backbone, which has a longer text encoder, using only f_theta (trained on Intent-CC3M) without De-MINDS. The experimental results are in Table 1 to Table 4 as below, proving the effectiveness of our De-MINDS across robust backbones:

###### Table 1: Comparative Results on the Fashion-IQ for Attribute Manipulation

| Model | Method       |          Dress          |         Shirt          |         Toptee         |         Average         |
| ----- | ------------ | :---------------------: | :--------------------: | :--------------------: | :---------------------: |
|       |              |      R@10    R@50       |      R@10    R@50      |      R@10    R@50      |      R@10    R@50       |
| ViT-L | Long-CLIP    |     21.23     44.03     |     27.96    46.18     |    30.27     51.04     |     26.40     47.08     |
|       | De-MINDS     |     25.20     48.70     |     31.00    51.20     |    32.90     55.70     |     29.70     51.90     |
| ViT-H | LinCIR       |     29.80     52.11     |     36.90    57.75     |     42.07    62.52     |     36.26     57.46     |
|       | De-MINDS     |     31.66     56.29     |     38.52    59.27     |     44.37    64.56     |     38.18     60.04     |
| ViT-G | CIReVL       |     27.07     49.53     |     26.89    45.58     |     29.32    49.97     |     25.56     46.23     |
|       | LinCIR       |     38.08     60.88     |     46.76    65.11     |     50.48    71.09     |     45.11     65.69     |
|       | **De-MINDS** | **39.74**     **62.27** | **47.88**    **67.53** | **53.42**    **73.42** | **46.91**     **67.74** |

###### Table 2: Comparative Results on the CIRR for Object Manipulation

| Model | Method       |    R@1    |    R@5    |   R@10    |
| ----- | :----------- | :-------: | :-------: | :-------: |
| ViT-L | Long-CLIP    |   24.37   |   52.77   |   66.84   |
|       | De-MINDS     |   27.30   |   57.00   |   71.30   |
| ViT-H | LinCIR       |   33.83   |   63.52   |   75.35   |
|       | De-MINDS     |   35.57   |   65.82   |   76.97   |
| ViT-G | CIReVL       |   34.65   |   64.29   |   75.06   |
|       | LinCIR       |   35.25   |   64.72   |   76.05   |
|       | **De-MINDS** | **37.62** | **66.85** | **78.14** |

###### Table 3: Comparative Results on the ImageNet for Domain Conversion

| Model | Method       |        Cartoon        |        Origami        |          Toy          |       Sculpture       |        Average        |
| ----- | ------------ | :-------------------: | :-------------------: | :-------------------: | :-------------------: | :-------------------: |
|       |              |     R@10    R@50      |     R@10    R@50      |     R@10    R@50      |     R@10    R@50      |     R@10    R@50      |
| ViT-L | Long-CLIP    |    8.9       23.7     |     15.8    26.1      |     9.4      22.8     |     10.9     25.7     |     11.3     24.6     |
|       | De-MINDS     |     13.3     31.2     |     20.3     34.5     |     14.7     31.7     |     16.5     34.7     |     16.2     33.0     |
| ViT-H | De-MINDS     |     14.7     32.2     |     21.7     35.8     |     15.9     33.0     |     17.6     35.7     |     17.5    34.2      |
| ViT-G | **De-MINDS** | **15.4**     **34.1** | **23.2**     **36.7** | **16.7**     **36.5** | **18.5**     **36.9** | **18.5**     **36.1** |

###### Table 4: Comparative Results  on the COCO for Object Composition

| Model | Method       |   R@1    |   R@5    |   R@10   |
| ----- | ------------ | :------: | :------: | :------: |
| ViT-L | Long-CLIP    |   12.7   |   26.3   |   36.4   |
|       | De-MINDS     |   15.7   |   33.2   |   44.1   |
| ViT-H | De-MINDS     |   16.9   |   34.7   |   45.6   |
| ViT-G | **De-MINDS** | **18.0** | **35.9** | **46.8** |

We observe significant performance improvement ranging from 1.89% to 2.25% with ViT-H and 1.93% to 2.20% using ViT- G over existing SoTA methods, proving the generalizability of our approach with different robust backbones. Moreover, utilizing the Longer text encoder (i.e., LongCLIP) as a backbone shows a performance decrease compared to De-MINDS by 4.06% to 6.65%. This further proves the effectiveness of our proposed framework on ZS-CIR tasks. We appreciate the reviewers' suggestions and will add these results to our revised manuscript.

**References**

[1] Saito, Kuniaki, et al. Pic2word: Mapping pictures to words for zero-shot composed image retrieval, CVPR 2023.

[2] Tang, Yuanmin, et al. Context-I2W: Mapping Images to Context-dependent Words for Accurate Zero-Shot Composed Image Retrieval,  AAAI 2024.

[3] Gabriel Ilharco, et al.  Openclip 2021.

[4] Zhang, Beichen, et al. Long-clip: Unlocking the long-text capability of clip, ECCV 2024.

---

### Decision · Program_Chairs · 2024-09-25

**Decision:**

Reject

**Comment:**

The paper presents De-MINDS, a framework designed to improve zero-shot composed image retrieval by accurately understanding human manipulation intentions expressed in textual modifications. By leveraging a newly created intention-based dataset and a specialized mapping network, De-MINDS significantly enhances retrieval accuracy across multiple tasks, establishing new state-of-the-art results. The paper had received mixed opinions / ratings due to unclear articulation of the "intention" concept and concerns about the novelty of its model architecture, which uses a common Q-Former approach without proper citations. Additionally, inadequate demonstration of the model's superiority in capturing "intention" weaken overall impact of the proposal. After considering the paper, reviews, rebuttal, and discussions, the AC agrees that the model's novelty is incremental and the idea of intention could be considered as a form of prompt augmentation / engineering, which is well explored in image retrieval research. Given these issues, the AC recommends rejecting the paper and advises the authors to address the reviewers' comments and consider resubmitting the revised work to a future conference.